# HEBBGATE: LOCAL REWARD-MODULATED GATING FOR CONTINUAL LEARNING

## ABSTRACT

Neural networks that learn continually should acquire new tasks without revisiting old data and with small per-task overhead. In parameter-isolation CL, existing approaches typically learn dense task masks via backpropagation, which couples mask learning to the backbone optimiser, adds training compute, and inflates memory with extra mask parameters. We introduce HebbGate, a parameter-isolation method for continual learning that uses local, reward-modulated gates in place of backpropagated masks. Crucially, each task adds just one scalar per channel (not per weight), keeping memory growth tiny and the masks interpretable. A utilisation penalty discourages reuse of over-popular channels, and a $\kappa$-decay capacity warm-up lets new tasks explore larger masks before annealing to the target sparsity, mitigating order bias and improving forward transfer. On CIFAR-100, Tiny-ImageNet-200, and ImageNet-100 with ResNet-18, HebbGate achieves best-known exemplar-free Class-IL final accuracy $A_{\text{last}}$ while a variant with task-specific BatchNorm further improves both $A_{\text{last}}$ and incremental accuracy $A_{\text{inc}}$ at the cost of only two additional scalars per channel per task. Additional experiments on Permuted-MNIST, Split-CIFAR-10, and lower-capacity backbones confirm that HebbGate's gains extend beyond a single architecture or dataset. Overall, HebbGate offers a lightweight, transparent alternative for exemplar-free, single-head continual learning (*Code to be released with paper acceptance*).

## 1 INTRODUCTION

Deep neural networks routinely top leaderboards in vision, speech, and language when the entire dataset is known in advance and large-scale training can be run once in a data center. Classic examples such as, ImageNet-pretrained ResNets and Vision Transformers, leverage millions of labelled images, powerful GPUs/TPUs, and many epochs of stochastic-gradient descent to optimise a single, fixed objective He et al. (2016); Dosovitskiy et al. (2021). In the wild, however, models embedded in drones, augmented-reality glasses, or smart-home cameras face a very different reality: new classes, environments, or user preferences arrive sequentially, bandwidth is limited, and limited resources rule out retraining from scratch each time the world changes. *Continual learning* (CL) tackles this streaming scenario by updating one network online instead of building a fresh model offline after every change. The central obstacle in CL is *catastrophic forgetting*, where gradient updates for the current task overwrite weights that were crucial for previous ones, causing a precipitous drop in past accuracy McCloskey & Cohen (1989); Goodfellow et al. (2013). The challenge peaks in *cold-start class-incremental learning* (Class-IL), where the learner receives disjoint class sets, stores no past data, and must predict over *all* classes at inference time Masana et al. (2022). Existing CL approaches (Section *Related Work*) span replay, synaptic regularisation, dynamic expansion, and parameter isolation. Within isolation, masks are still (i) learned by backpropagation, (ii) stored as dense vectors, and (iii) prone to order bias that lets early tasks monopolise low-level channels.

We propose HEBBGATE, a *local* and *transparent* gating mechanism that keeps the strong retention of parameter isolation while reducing its gradient and compute overhead. Each channel stores a single scalar gate; a reward-modulated Hebbian rule updates that scalar using only the current activation magnitude and a signed batch-level margin reward. A capacity-aware factor down-weights heavily-used neurons, biasing learning towards underutilized units, so new tasks tend to recruit fresh filters, keeping inter-task overlap close to the target sparsity $\kappa$. The resulting system is exemplar-free,

converges faster than gradient-based methods, has low memory overheads. Our main contributions are the following:

1. **Local, reward-driven Hebbian gating:** A lightweight Hebbian rule that updates *one scalar per neuron* based on a three-factor signal: (i) activation energy, (ii) a signed margin reward $r$, and (iii) usage-aware scaling. No gradients ever flow through the gates, decoupling mask learning from backbone optimization and reducing per-task training time.

2. **Adaptive capacity control:** A utilization-aware penalty that steers new tasks toward under-utilised channels and down-weights neurons that were frequently selected by previous tasks. To complement this, we introduce a novel $\kappa$-*decay warm-up* schedule: new tasks initially explore a larger subset of units, and gradually anneal to a target sparsity $\kappa$. This balances plasticity and stability, prevents early tasks from monopolising shallow layers, and improves forward transfer while keeping inter-task overlap near the budget.

3. **Single-head Class-IL via fused parallel gating:** We eliminate any extra routing pass or auxiliary classifier by stacking each test sample across all $T$ task-IDs and performing one fused forward pass. This method uses only channel-wise multiplication for the learned masks, selecting the best subnetwork for classification.

We benchmark on 10-task Class-IL splits of CIFAR-100, Tiny-ImageNet-200, and ImageNet-100 with ResNet-18, as well as on Split-CIFAR-10 and Permuted-MNIST with lower-capacity backbones, and show that HEBBGATE achieves best-known exemplar-free Class-IL final accuracy with moderate forgetting, positive forward transfer, and negligible memory overhead.

## 2 RELATED WORK

We consider *offline, class-incremental* learning with a single shared classifier (no task label at test time). A long line of work, including Experience Replay (ER) Chaudhry et al. (2019) and Deep Generative Replay (DGR) Shin et al. (2017), tackles the catastrophic forgetting problem by storing or synthesising past samples. More recent approaches, such as CCIL Wang et al. (2025), use data-free replay based on feature inversion and consistency constraints, combined with debiased classifiers. Replay consistently yields strong retention but incurs linear memory growth (or a trained generator) and raises privacy concerns, making it better viewed as a complementary enhancement rather than a primary evaluation axis.

Regularization and distillation methods are exemplar-free approaches that constrain parameter drift or outputs to preserve past knowledge. Weight-importance penalties such as, EWC Kirkpatrick et al. (2017), SI Zenke et al. (2017), and MAS Aljundi et al. (2018), restrict updates along critical weights. Functional distillation (LwF/LwM Li & Hoiem (2017)) matches logits from past models, while FeTrIL Petit et al. (2023) uses distillation in feature space by maintaining class prototypes. Several recent methods operate primarily in feature space. SSRE Zhu et al. (2022) reorganises network structure with self-supervised objectives to expand representation space without exemplars. FeCAM Goswami et al. (2023) freezes the backbone after the first task and learns incrementally in feature space by maintaining covariance-aware class prototypes and using an anisotropic Mahalanobis classifier. Elastic Feature Consolidation (EFC) Magistri et al. (2024) tackles cold-start incremental learning by estimating an empirical feature matrix per task and regularising drift along high-variance feature directions. Learnable Drift Compensation (LDC) Gomez-Villa et al. (2024) estimates the semantic drift of a moving backbone and learns a projector that maps old-task prototypes into the current feature space. Class-wise decorrelation (CwD) Konishi et al. (2023) introduces a covariance-shaping objective that encourages uniformly spread class features and improves downstream incremental performance. Although memory-bounded, these methods are computationally heavy. They require repeated backpropagation through auxiliary penalties, storage of importance matrices or teacher models, much longer training schedules per task, and remain tightly coupled to the backbone optimiser.

A second line of work is Parameter Isolation approaches, which embeds multiple subnetworks inside a shared backbone, structurally decoupling gradient flows across tasks. PackNet prunes and re-trains weights per task Mallya & Lazebnik (2018); Piggyback Mallya et al. (2018) learns binary masks over a frozen base; and Supermasks and SupSup share weights in superposition Wortsman et al. (2020). Progressive Networks (PNNs) widens capacity with lateral adapters Rusu et al.

(2016). Differentiable gating schemes such as HAT Serra et al. (2018) and CCG Abati et al. (2020) learn near-binary masks with sparsity penalties, but introduce thousands of extra gating parameters and demand long training schedules. Other isolation variants, e.g. CDG Masse et al. (2018) and ADN Iyer et al. (2022), rely on task cues at inference and thus fall outside the single-head Class-IL setting. These methods can mitigate forgetting effectively because of capacity partitioning and isolation, which leads to reduced gradient interference (hence forgetting) but also restricts feature reuse, leaving later tasks unable to exploit high-utility filters and harming forward and backward transfer.

**Research gaps.** Two structural gaps remain. Feature-space approaches such as FeTrIL, SSRE, FeCAM, EFC, and LDC preserve past knowledge via stored prototypes or per-task statistics, but rely on global backpropagation through a shared backbone and offer no mechanism for task-specific control over parameter usage. Parameter-isolation methods instead allocate disjoint subnetworks, yet existing variants learn masks through gradient-based optimisation, rely on large mask tensors, and provide no regulation of cross-task usage. As a result, early tasks often monopolise high-utility channels (especially in shallow layers), leaving later tasks with limited capacity to explore or reuse features. A further limitation is the use of fixed sparsity budgets where overly sparse masks limit positive feature sharing, while overly dense masks increase interference. No current exemplar-free method resolves this isolation-plasticity trade-off with a principled, local update rule.

HEBBGATE addresses these gaps by learning task-specific sparse subnetworks using a local, reward-modulated Hebbian update on a single scalar per channel, without backpropagation through gates, without prototype storage, and with explicit usage-aware penalisation. A $\kappa$–decay schedule enables early exploration and later consolidation, preventing early tasks from saturating capacity while leaving space and trained features for later tasks enabling forward transfer.

## 3 INCREMENTAL-LEARNING SETUP

The learner observes a sequence of $T$ classification tasks $\{\mathbb{T}_t\}_{t=1}^T$. Task $t$ provides a training set $\mathbb{D}_t^{\text{train}} = \{(x_n^{(t)}, y_n^{(t)})\}_{n=1}^{N_t}$ with $x_n^{(t)} \in \mathbb{X}_t$ and $y_n^{(t)} \in \mathbb{C}_t$, where the class sets are disjoint, $\mathbb{C}_t \cap \mathbb{C}_{t'} = \varnothing$ for $t \neq t'$. After training on $\mathbb{T}_t$ the data are discarded; the network then receives $\mathbb{T}_{t+1}$. We evaluate two standard regimes. *Task-Incremental Learning* (Task-IL) supplies the task index $t$ at inference time, permitting task-specific heads or masks. *Class-Incremental Learning* (Class-IL) provides no task oracle; the model must predict over the union $\mathbb{C} = \bigcup_{t=1}^T \mathbb{C}_t$ through a *single* shared soft-max, making it the most stringent exemplar-free protocol.

**Design desiderata.** A continual-learning system should (i) *mitigate forgetting* by retaining high accuracy on past tasks; (ii) *avoid replay* by requiring no stored or generated data; (iii) *bound resource growth* so that extra weights and FLOPs increase sub-linearly with the number of tasks $T$ (one scalar per channel/neuron), making the growth negligible relative to the total number of weights; and (iv) *promote transfer* by delivering positive forward and backward transfer: new tasks gain from previously learned features, while earlier tasks improve as the system is refined on later ones.

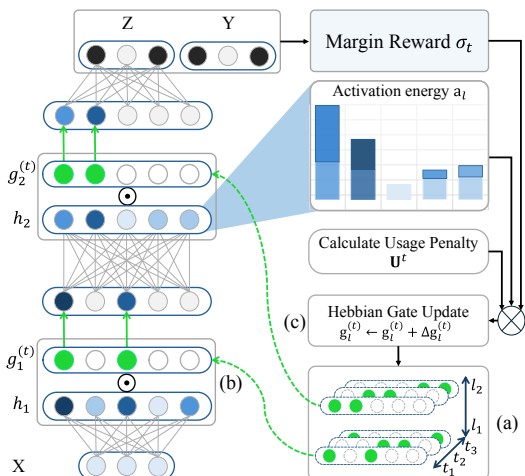

## 4 HEBBGATE

*HebbGate* is a local, interpretable utilization-aware gating mechanism designed to satisfy the four criteria without gradient-based mask learning or replay.

Figure 1: HebbGate pipeline: (a) Gate bank stores one scalar per channel/task; (b) Masking activations with top-$\kappa$ entries (green); (c) Local Hebbian update combining activation energy, margin reward, and usage.

**Pipeline overview.** The same three-factor update is repeated for every mini-batch, after which the task's gates are *frozen*. Figure 1 sketches the three recurring stages executed for every task $t$. Each gated layer $\ell$ stores a vector $\mathbf{g}_\ell^{(i)} \in [0,1]^{d_\ell}$ for every past task $i \le t$. The vector of dimensionality $d_\ell$ holds one scalar *gate* per channel (or neuron) and occupies fixed memory footprint. When an input $x$ from task $t$ arrives, the model multiplies the backbone activations by a top-$\kappa$ mask $M_\kappa(\mathbf{g}_\ell^{(t)}) \in [0,1]^{d_\ell}$, keeping only the $\kappa d_\ell$ largest gates (green circles in the figure) and zeroing the rest (grey). The shared classifier head produces logits $\mathbf{z}(x)$ over the cumulative class set $\mathbb{C}$. After the weight-update step we collect three local signals: *(i)* channel energy $a_{\ell,j} = \|h_{\ell,j}\|_2^2$, where $h_{\ell,j}$ are layer activations, *(ii)* a signed batch-level margin reward $r_t \in [-1,1]$, and *(iii)* the historical usage $u_{\ell,j} \in [0,1]$ of the same channel. A lightweight Hebbian rule $\Delta g_{\ell,j} = \eta\,(a_{\ell,j} - \bar{a}_\ell)\,r_t\,(1 - u_{\ell,j})^\gamma$ updates the gate scalars *without backpropagation*. Usage is accumulated online, and a linear $\kappa$-decay schedule gradually tightens the mask budget from an initial $\kappa_{\text{start}}$ down to the target $\kappa_{\min} = 1/T$.

## 4.1 GATED LAYERS AND DYNAMIC MASKING

For every layer $\ell = 1, \ldots, L$ we allocate a *task-specific gating vector*

$$\mathbf{g}_\ell^{(t)} \in [0,1]^{d_\ell}, \qquad t = 1, \ldots, T, \tag{1}$$

whose components control which of the $d_\ell$ neurons (channels for CNNs) are active when solving task $t$. These vectors are updated in tandem with the model weights, but via a local Hebbian update rule instead of backpropagation. At run time the latent gate is converted into a binary mask.

We wrap every *Convolutional* and *Linear* layer inside a *Gated-Conv / Gated-Linear* module that augments the weight tensor with a *gate bank*. For each task $t \le T$ and each layer $\ell = 1, \ldots, L$ the bank stores a gating vector

$$\mathbf{g}_\ell^{(t)} \in \mathbb{R}^{d_\ell}, \tag{2}$$

one scalar per channel (for Gated-Conv) or neuron (for Gated-Linear). The backbone weights $\mathbf{W}_\ell$ are shared across tasks and only the gate scalars differ. At run time a latent gate is transformed into a binary (or soft) mask by the top-$k$ operator

$$M_\kappa(\mathbf{g}_\ell^{(t)}) = \text{topk}\big(\mathbf{g}_\ell^{(t)}, k = \lceil \kappa d_\ell \rceil\big) \in \{0,1\}^{d_\ell}, \tag{3a}$$

$$\tilde{M}_\kappa(\mathbf{g}_\ell^{(t)}) = \mathbf{g}_\ell^{(t)} \odot M_\kappa(\mathbf{g}_\ell^{(t)}) \quad \text{(soft variant).} \tag{3b}$$

Hard-masking (equation 3a) keeps the $\kappa d_\ell$ largest entries and zeros the rest, whereas soft-masking (equation 3b) preserves their magnitudes. Unless stated otherwise we report results with *hard* masks, which add only $\mathcal{O}(d_\ell)$ element-wise multiplies per layer.

**Utilization-aware gate initialization.** When task $t$ arrives we create a fresh gate vector by biased sampling:

$$g_{\ell,j}^{(t)} \sim \begin{cases} \mathcal{U}(0.4, 0.7) & \text{if } u_{\ell,j} < \theta, \\ \mathcal{U}(0, \varepsilon_{\ell,j}) & \text{otherwise,} \end{cases}$$

where $U(a,b)$ denotes the uniform distribution in $[a,b]$, $u_{\ell,j}$ is the historical usage of channel $j$ (defined in Sec. 4.3), $\theta = 1/I$ where $I$ is the number of tasks that completed training and $\varepsilon_{\ell,j} = 1 - u_{\ell,j}$ in all experiments. Unused channels therefore start high and have a better opportunity to be in the top-$k$, whereas heavily reused channels start near zero.

**Sparsity $\kappa$ warm-up.** Fixing the sparsity budget from the first mini-batch hampers early tasks, especially in narrow first stages (64 channels in ResNet-18). We therefore begin each task with a larger mask fraction $\kappa_{\text{start}}$ and linearly decay it to the minimum $\kappa_{\min} = 1/T$ over $E_t$ training epochs. Fig. 2 visualizes the utilization-aware initialization along with the $\kappa$-decay warm-up process. The warm-up lets SGD explore a larger parameter sub-space, improving forward transfer

$$\kappa_t(e) = \max\big(\kappa_{\min},\ \kappa_{\text{start}} - \tfrac{e}{E_t}(\kappa_{\text{start}} - \kappa_{\min})\big). \tag{4}$$

**Implications.** *(i)* Memory grows only linearly in $d_\ell$ and *sub-linearly* in $T$ thanks to the shrinking $\kappa_t$. *(ii)* Hard masking gives strict parameter isolation, yet soft masks are available for tasks that benefit from finer control. *(iii)* Usage-aware initialization plus $\kappa$-decay biases each new task toward fresh capacity, keeping inter-task overlap close to $\kappa$ while still permitting beneficial reuse when a channel truly helps multiple tasks enabling positive forward transfer.

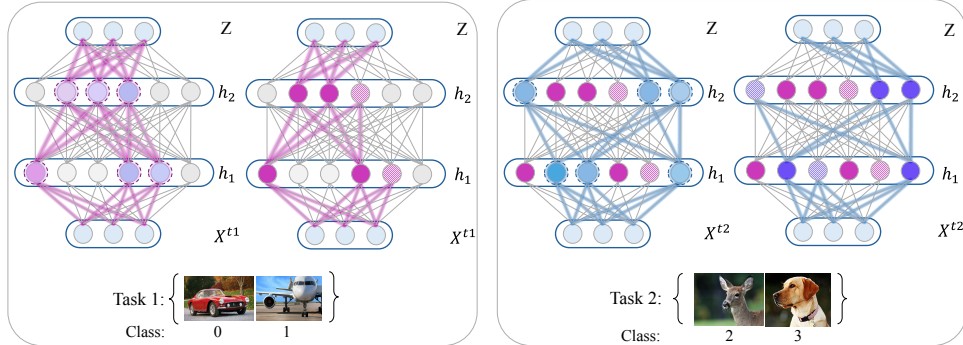

Figure 2: **Utility-aware gate initialization and $\kappa$-decay warm-up.** *Left Square:* at the **start** of task $t=1$ (left network) each layer activates a generous mask budget ($\kappa_{\text{start}}$, dashed purple). A linear warm-up schedule shrinks the budget to $\kappa_{\min} = 1/T$ as training proceeds, and the top-$k$ entries are frozen (purple) when the task finishes (right network). *Right Square:* beginning task $t=2$ (left network) the new gate vector is *biased* toward channels that were *unused* so far (dashed circles)

## 4.2 FORWARD PASS

We now embed the task-conditioned masks from Sec. 4.1 into the network computation. Given the previous activation $\mathbf{h}_{\ell-1} \in \mathbb{R}^{B \times d_{\ell-1}}$ ($B$ = batch size) the masked forward computation is

$$\mathbf{h}_\ell = \phi\big(\mathbf{h}_{\ell-1}\mathbf{W}_\ell\big) \odot M_\kappa\big(\mathbf{g}_\ell^{(t)}\big), \tag{5}$$

where $\phi$ is the ReLU. For a feed-forward backbone with $L$ hidden layers the recursion of equation 5 propagates activations through the *task-specific subnetwork*, and $\mathbf{W}_\ell \in \mathbb{R}^{d_{\ell-1} \times d_\ell}$ are the shared weights and $\phi(\cdot)$ is ReLU throughout. The element-wise product with $\mathbf{M}_\kappa$ selects the $k_\ell$ neurons assigned to task $t$, leaving the remaining $d_\ell - k_\ell$ units *exactly zero*, and hence both forward and backward passes are restricted to the active subset. The final representation $\mathbf{h}_L^{(t)}$ feeds an *ungated* affine layer $\mathbf{W}_{L+1} \in \mathbb{R}^{d_L \times \mathcal{C}_t}$ that outputs logits $\mathbf{z}^{(t)} = \mathbf{h}_L^{(t)}\mathbf{W}_{L+1}$. Keeping the head ungated avoids accidental removal of class logits and simplifies continual evaluation. Given one-hot labels $\mathbf{Y}^{(t)} \in \{0, 1\}^{B \times C}$, training optimises the cross-entropy loss

$$\mathcal{L}_{\text{CE}} = -\frac{1}{B}\sum_{b=1}^{B} \mathbf{Y}^{(t)}[b] \cdot \log\big(\text{softmax}(\mathbf{z}^{(t)}[b])\big), \tag{6}$$

with respect to all weights $\{\mathbf{W}_\ell\}_{\ell=1}^{L+1}$, while the gate vectors $\{\mathbf{g}_\ell^{(t)}\}$ are held fixed during the back-propagation step and updated separately by the local Hebbian rule (discussed below). Because $\mathbf{M}_\kappa$ is binary, each task traverses a strict, in the case of hard masking, subnetwork of the backbone. Tasks can overlap − sharing neurons where beneficial − or isolate themselves completely when interference outweighs reuse. The degree of overlap emerges dynamically from the gated learning rule and is neither imposed nor tuned by hand.

## 4.3 REWARD–MODULATED HEBBIAN PLASTICITY

After each weight-update step the latent task gates $\{\mathbf{g}_\ell^{(t)}\}_{\ell=1}^{L-1}$ are adapted by a purely local, three-factor Hebbian rule that couples (i) neuronal co-activity, (ii) a task-level reinforcement signal and (iii) a capacity-control usage penalty that prevents recurrent reuse of the same units. Let $\mathbf{z}_b \in \mathbb{R}^{|\mathcal{C}_t|}$ be the logits for sample $b$ and $y_b$ its ground-truth label. We compute a smooth, bounded **margin** reward signal by averaging the hyperbolic tangent

$$r_t = \frac{1}{B}\sum_{b=1}^{B} \tanh\big(z_{b,y_b} - \max_{c \neq y_b} z_{b,c}\big) \in (-1, 1), \tag{7}$$

---

**Algorithm 1** HEBBGATE training for task $t$

---

1: **Input:** task data $\mathbb{D}_t$, current backbone $\{\mathbf{W}_\ell\}$, gate bank $\mathbb{G}$
2: Initialise new gates $\left\{\mathbf{g}_\ell^{(t)}\right\}_{\ell=1}^L$ with utility–aware sampling (Sec. 4.1)
3: **for** epoch $e = 1$ **to** $E_{\mathrm{P}}$ **do**
4:    **for** mini-batch $(x, y) \sim \mathbb{D}_t$ **do**
5:      *// zero-clamp outside top-$\kappa_t(e)$*
6:      $\mathbf{g}_\ell^{(t)} \leftarrow M_{\kappa_t(e)}(\mathbf{g}_\ell^{(t)})$    **for all** $\ell$
7:      Forward $\longrightarrow$ logits $\mathbf{z}$, store activations
8:      Back-prop cross-entropy, update $\{\mathbf{W}_\ell\}$
9:      Compute reward $r_t$ equation 7
10:     Hebbian gate update $\mathbf{g}_\ell^{(t)}$ via equation 10 **for all** $\ell$
11:     **clip** $g_\ell^{(t)} \leftarrow \mathrm{clip}\left(g_\ell^{(t)}, 0, 1\right)$ *// post-update clipping*
12:    **end for**
13: **end for**
14: **for** epoch $e = 1$ **to** $E_{\mathrm{W}}$ **do**
15:    Freeze gates, train $\{\mathbf{W}_\ell\}$ only
16: **end for**
17: $g_\ell^{(t)} \leftarrow M_{\kappa_{\min}}(g_\ell^{(t)})$; update usage $u_{\ell,j}$ *// binary freeze*

---

which is positive for confidently correct batches and negative for confidently wrong ones [1]. For every channel $j$ in layer $\ell$, we measure its activation energy $a_{\ell,j} \triangleq \|h_{\ell,j}\|_2^2$, computed as:

$$a_{\ell,j} = \begin{cases} \frac{1}{B} \sum_b \|h_{\ell,j}^{(b)}\|_2^2 & \text{(MLP)}, \\ \frac{1}{B} \sum_b \frac{1}{HW} \sum_{x,y} (h_{\ell,j}^{(b)}[s, w])^2 & \text{(CNN)}, \end{cases} \tag{8}$$

where $s \in H, w \in W$ are spatial dimensions. We centre the energy by its layer mean $\bar{a}_\ell = \frac{1}{d_\ell} \sum_j a_{\ell,j}$ so that only channels stronger than average receive a positive boost. Historical utilization of channel/neuron $j$ up to task $i$ (excluding task $t$) is given by

$$u_{\ell,j}^{(t)} := \frac{1}{t-1} \sum_{i=1}^{t-1} g_{\ell,j}^{(i)} \in [0, 1],$$

denoting how often channel $j$ has entered the top-$\kappa$ mask of previous tasks (Section *Gated layers and dynamic masking*). A power-law penalty suppresses highly reused channels

$$s_{\ell,j} = (1 - u_{\ell,j})^\gamma. \tag{9}$$

Combining equation 7–equation 9 yields the *reward–modulated Hebbian gate update rule*

$$\mathbf{g}_\ell^{(t)} \leftarrow \mathbf{g}_\ell^{(t)} + \boxed{\eta_g \, (a_{\ell,j} - \bar{a}_\ell) \, r_t \, (1 - u_{\ell,j})^\gamma,} \tag{10}$$

with learning rate $\eta_g$. The rule is fully local, meaning every gate uses only its own activation energy, a global scalar reward, and its historical usage, while no gradients flow through $\mathbf{g}$. Channels that fire strongly and help the batch (large positive $(a - \bar{a})r_t$) increase their gate value unless they are already heavily used, while unhelpful or overused channels are suppressed.

**Implications.** The update is computationally efficient, with complexity $\mathcal{O}(d_\ell)$, and can be executed on-device after each mini-batch. Usage scaling naturally maintains inter-task overlap near the target sparsity $\kappa$, eliminating the need for an explicit sparsity loss. Moreover, since masks are never backpropagated and training schedules are significantly shorter.

### 4.4 TRAINING LOOP AND PHASE SCHEDULE

We process the $T$ tasks sequentially. For every task $t$ the optimization is split into two short **phases**: (A) **Parallel phase**; (B) **Weights phase**. The whole protocol is summarised in Algorithm 1. During

---

[1] A binary $\pm 1$ reward based on batch accuracy is analysed in Appendix D; the smooth margin variant in equation 7 proved more stable.

parallel phase gates and backbone weights are updated together for $E_{\mathrm{P}}$ epochs. Each mini-batch performs *(i)* SGD on $\{\mathbf{W}_\ell\}$ with cross-entropy loss, *(ii)* the Hebbian gate update (equation 10). Before the forward pass we zero-clamp every gate outside the current top-$\kappa_t(e)$ entries (equation 3a), guaranteeing the effective mask size never exceeds the epoch-wise budget $\kappa_t(e)$ of equation 4. After every Hebbian update in equation 10, we clip gate values element-wise to the unit interval, $g_\ell^{(t)} \leftarrow \mathrm{clip}(g_\ell^{(t)}, 0, 1)$. In Weights phase gates are frozen; only $\{\mathbf{W}_\ell\}$ receive $E_{\mathrm{W}}$ fine-tuning epochs with a reduced learning rate. At the end of phase (B) we permanently freeze the gates of task $t$:

$$\mathbf{g}_\ell^{(t)} := M_{\kappa_{\min}}(\mathbf{g}_\ell^{(t)}), \qquad \ell = 1, \ldots, L, \tag{11}$$

i.e. keep the final top-$\kappa_{\min}$ entries and set the rest to zero.[2] The binary vector is then accumulated into the usage statistic $u_{\ell,j} \leftarrow \frac{t-1}{t} u_{\ell,j} + \frac{1}{t} g_{\ell,j}^{(t)}$, which feeds the scaling factor (equation 9).

### 4.5 CLASS-INCREMENTAL INFERENCE VIA PARALLEL GATING

In the Class-IL protocol, the task identity is unknown at test time. We evaluate all $T$ gated subnetworks and select the most confident one, using either a fused (batched) or a sequential computation. Let $x \in \mathcal{X}$ be a test sample, $C$ the number of classes in the shared head, and $f(x; t) \in \mathbb{R}^C$ the logits produced by the subnetwork gated by $\mathbf{g}^{(t)}$ (i.e., mask $M^{(t)}$ applied at each layer of width $d_\ell$). We define:

$$\textbf{Parallel/fused:} \quad \widetilde{Z}(x) = [\, f(x; 1), \ldots, f(x; T) \,] \in \mathbb{R}^{T \times C}, \tag{12a}$$

$$\textbf{Sequential:} \quad \hat{Z}_t(x) = f(x; t) \text{ for } t = 1, \ldots, T, \tag{12b}$$

In evaluation mode, equation 12a and equation 12b produce the *same* set $\{f(x; 1), \ldots, f(x; T)\}$ and thus identical predictions. The fused path executes a single $T$-way pass and adds $O(T \sum_\ell d_\ell)$ element-wise multiplies for masks (no extra learnable parameters), while the sequential path runs $T$ forwards in series (useful under tight memory).

**Task score and selection.** We score each task $t$ and pick

$$\hat{t}(x) = \arg\max_t s_t(x), \qquad \hat{y}(x) = \arg\max_c f(x; \hat{t})_c.$$

Different masks induce task-dependent logit scales, so raw confidence $s_t(x) = \max_c \mathrm{softmax}(f(x; t))_c$ is not directly comparable across $t$. We therefore apply *z-normalization* with moments estimated once on a small calibration split:

$$\tilde{s}_t(x) = \frac{s_t(x) - \mu_t}{\sigma_t + \varepsilon}, \qquad \hat{t}(x) = \arg\max_t \tilde{s}_t(x).$$

For mini-batches we average $\tilde{s}_t$ over samples before the $\arg\max$; this matches the fused implementation. This standardization removes nuisance scale differences across tasks and consistently improves Class-IL accuracy without any learned calibrators.[3]

## 5 EXPERIMENTS

We evaluate the position of HebbGate as an exemplar-free alternative to replay buffers and heavy regularisers, combining fast local updates, minimal memory cost and strong accuracy[4].

**Benchmarks and protocols.** Our primary evaluation is single-head Class-IL (no task labels at test time) on CIFAR-100, Tiny-ImageNet-200, and ImageNet-100, each split into $T=10$ disjoint class-ordered tasks. We report the average accuracy over all tasks, $A_{\mathrm{last}}$, and the average accuracy over all intermediate steps, $A_{\mathrm{inc}}$. We also report backward transfer (BWT), forgetting, and a forward-transfer proxy $F$, (definitions in App. A.3). Task-IL metrics are used only as secondary analysis.

---

[2]Using the binary version avoids ambiguity when computing usage.

[3]Calibration protocol and estimation of $(\mu_t, \sigma_t)$ are detailed in App. §A.4.

[4]**Cold-start accuracy** (or zero-shot): the test accuracy obtained on a previously unseen task *immediately* after the model has finished training on the earlier tasks

Table 1: Exemplar-free single-head Class-IL with $T{=}10$ tasks on CIFAR-100, Tiny-ImageNet-200, and ImageNet-100 using ResNet-18. We report final accuracy $A_{\text{last}}$ and incremental accuracy $A_{\text{inc}}$ (mean $\pm$ std over 5 seeds). Best results are in **bold**, second-best within std range underlined.

| Method | CIFAR-100 | | Tiny-ImageNet | | ImageNet-100 | |
|---|---|---|---|---|---|---|
| | $A_{\text{last}}$ | $A_{\text{inc}}$ | $A_{\text{last}}$ | $A_{\text{inc}}$ | $A_{\text{last}}$ | $A_{\text{inc}}$ |
| SSRE | $30.4 \pm 0.7$ | $47.3 \pm 1.9$ | $22.9 \pm 1.0$ | $38.8 \pm 2.0$ | $25.4 \pm 1.2$ | $43.8 \pm 1.1$ |
| LwF | $32.8 \pm 3.1$ | $53.9 \pm 1.7$ | $26.1 \pm 1.3$ | $45.1 \pm 0.9$ | $37.7 \pm 2.5$ | $56.4 \pm 1.0$ |
| PASS | $30.5 \pm 1.0$ | $47.9 \pm 1.9$ | $24.1 \pm 0.5$ | $39.3 \pm 0.9$ | $26.4 \pm 1.3$ | $45.7 \pm 0.2$ |
| EWC | $31.2 \pm 2.9$ | $49.1 \pm 1.3$ | $8.0 \pm 0.3$ | $24.0 \pm 0.5$ | $24.6 \pm 4.1$ | $39.4 \pm 3.1$ |
| FeTrIL | $34.9 \pm 0.5$ | $51.2 \pm 1.1$ | $31.0 \pm 0.9$ | $45.6 \pm 1.7$ | $36.2 \pm 1.2$ | $52.6 \pm 0.6$ |
| FeCAM | $33.1 \pm 0.9$ | $48.1 \pm 1.3$ | $24.9 \pm 0.5$ | $38.6 \pm 1.3$ | $42.4 \pm 0.9$ | $57.9 \pm 1.5$ |
| EFC | $43.6 \pm 0.7$ | $58.6 \pm 0.9$ | $\underline{34.1 \pm 0.8}$ | $\underline{48.0 \pm 0.6}$ | $47.4 \pm 1.4$ | $59.9 \pm 1.4$ |
| LDC | $45.4 \pm 2.8$ | $59.5 \pm 3.9$ | $\underline{34.2 \pm 0.7}$ | $46.8 \pm 1.1$ | $\underline{51.4 \pm 1.2}$ | $\mathbf{69.4 \pm 0.6}$ |
| CCIL | $43.7 \pm 0.6$ | $\underline{60.1 \pm 1.9}$ | $30.9 \pm 0.5$ | $45.3 \pm 0.7$ | $46.0 \pm 1.1$ | $62.7 \pm 1.1$ |
| **HebbGate-IN** | $58.8 \pm 0.9$ | $61.7 \pm 0.9$ | $34.1 \pm 1.2$ | $34.9 \pm 1.5$ | $\underline{55.3 \pm 1.6}$ | $54.5 \pm 2.0$ |
| **HebbGate-BN** | $\mathbf{71.8 \pm 0.8}$ | $\mathbf{74.8 \pm 0.6}$ | $\mathbf{48.5 \pm 0.3}$ | $\mathbf{51.1 \pm 0.8}$ | $\mathbf{64.3 \pm 1.6}$ | $\underline{63.7 \pm 0.5}$ |

**Architectures and normalization.** Unless otherwise stated, we use ResNet-18 He et al. (2016) as the main backbone for all three image benchmarks. For ablations and comparison with earlier parameter-isolation baselines we also use a 3-conv AlexNet and a two-layer MLP-2000 (Table 6). We avoid running-statistic leakage induced by shared BatchNorm using InstanceNorm or Group-Norm in the main configuration (**HebbGate-IN**), and use a single shared classifier head. Because HebbGate already allocates task-specific sparse subnetworks, a natural variant is to also maintain task-specific BatchNorm statistics for each subnetwork which yields the **HebbGate-BN** configuration. Inference remains task-agnostic since we evaluate all gated subnetworks with their own BN statistics aggregating their outputs exactly as in the IN/GN setting adding only two scalars per channel per task (running mean/variance). Extended reproducibility details, dataset splits, training schedules, and metric definitions are given in App. A.2–A.3.

## 5.1 COMPARISON WITH RELATED WORK

We compare HebbGate against strong *exemplar-free* baselines: (i) regularisation and distillation methods (EWC, LwF, PASS), (ii) feature-space methods based on prototypes or feature statistics (SSRE, FeTrIL, FeCAM, EFC, LDC), and (iii) data-free replay (CCIL). All methods use the same 10-task protocol with a shared ResNet-18 backbone and single-head Class-IL evaluation. Replay-based CCIL is included as a strong reference point, but it uses synthetic data, whereas HebbGate and the remaining methods do not store or synthesize exemplars.

Table 1 summarises the main Class-IL results. On CIFAR-100, the **HebbGate-IN** configuration (without per-task BatchNorm) already achieves the highest final accuracy $A_{\text{last}}$ among exemplar-free methods (58.8%), improving substantially over EFC (43.6%) and LDC (45.4%). The task-conditioned normalisation variant **HebbGate-BN** further increases both $A_{\text{last}}$ and $A_{\text{inc}}$ to 71.8% and 74.8%, respectively, indicating that a significant fraction of residual forgetting is mediated by shared normalisation statistics rather than parameter interference alone. On the more challenging Tiny-ImageNet-200, HebbGate-IN achieves final accuracy on par with the best exemplar-free baselines (34.1% vs. 34.2% for LDC and 34.1% for EFC, all within one standard deviation), while HebbGate-BN reaches 48.5% $A_{\text{last}}$ and 51.1% $A_{\text{inc}}$, clearly outperforming all baselines. On ImageNet-100, HebbGate-IN again achieves the highest $A_{\text{last}}$ among exemplar-free methods (55.3% vs. 51.4% for LDC), and HebbGate-BN pushes $A_{\text{last}}$ to 64.3% with competitive $A_{\text{inc}}$. We report **HebbGate-IN** as a natural reference point in the standard setting, which matches the assumptions made by most prior work. **HebbGate-BN** illustrates the additional headroom unlocked when task-conditioned normalisation is aligned with the gated subnetworks, using per-task BatchNorm statistics that integrate naturally with HebbGate's task-specific channel masks.

**Compute, memory, and inference complexity.** In terms of efficiency, HebbGate stores only per-channel gate scalars (and, in HebbGate-BN, per-task BN statistics), so its test-time parameter memory remains essentially equal to the base network: for ResNet-18 and $T{=}10$ tasks, the gate

Table 2: Test-time memory (FP32, $T{=}10$, batch=128) and inference latency on ResNet-18.

| Method | Extra per-task params | MB | ×base | ms/batch | ×HebbGate |
|---|---|---|---|---|---|
| HebbGate-IN (+BN) | gates (+BN stats) | $\approx 47.0$ | 1.0 | 2.16 | 1.0 |
| FeCAM / LDC | class prototypes / centroids | $\approx 47.0$ | 1.0 | – | – |
| EFC | none at test-time | 46.8 | 1.0 | 19.12 | 8.9 |
| EWC | Fisher diagonal (one / weight) | 93.6 | 2.0 | 32.87 | 15.2 |
| HAT | per-weight task masks | 467.6 | 10.0 | 42.97 | 19.9 |

Table 3: Backward transfer (BWT) and forward-transfer proxy $F$ for HebbGate in Class-IL.

| Setting | Dataset | BWT | $F$ |
|---|---|---|---|
| CIL | CIFAR-100 | $-4.35 \pm 0.72$ | $7.28 \pm 2.46$ |
| CIL | ImageNet-100 | $-5.48 \pm 1.50$ | $6.72 \pm 1.11$ |

bank adds $\sim 80\,\text{kB}$ and BN statistics add $\sim 160\,\text{kB}$. Feature-space methods such as FeCAM, LDC, and EFC accumulate one prototype or centroid vector per class and remain within $< 1\%$ of the base parameter count, while methods with per-weight statistics or masks (EWC, HAT) inflate memory by roughly $2\times$ and $10\times$, respectively (Table 2, App. A.8). At inference, HebbGate evaluates all task-conditioned subnetworks in a single fused forward pass by applying gates as channel-wise multipliers inside grouped convolutions. This adds only a small number of extra multiplies leading to per-batch latency close to the base model, while classical isolation and regularisation baselines (EFC, EWC, and HAT) that rely on per-weight statistics can be an order of magnitude slower.

**Task-IL.** On CIFAR-100, HebbGate matches SPG and PathNet while using a single shared head instead of task-aware multi-head inference. On Split-CIFAR-10/100 with AlexNet, HebbGate also outperforms prior parameter-isolation and regularisation baselines (App. A.5), showing that the gains are not restricted to high-capacity ResNet-18 models or a particular dataset.

Across CIFAR-100, Tiny-ImageNet-200, and ImageNet-100, HebbGate achieves the best-known exemplar-free Class-IL final accuracy $A_{\text{last}}$ while remaining close to backbone-level memory and compute, and without exemplars, distillation teachers, auxiliary penalties, or multi-head inference.

## 5.2 Ablation and Behaviour Analysis

We analyse HebbGate's behaviour through standard continual-learning metrics (backward transfer, forgetting, forward transfer) and a set of targeted ablations. Unless otherwise stated, we report ResNet-18 results; extended tables and per-task curves are given in App. A.6–A.7.

**Backward transfer and forgetting.** Following recent exemplar-free CL work Magistri et al. (2024); Goswami et al. (2023); Gomez-Villa et al. (2024), we measure backward transfer (BWT) and forgetting from the accuracy matrix $\mathcal{A}(t, i)$, where $\mathcal{A}(t, i)$ denotes the test accuracy on task $i$ after training task $t$. BWT is the average change in performance on past tasks after learning new ones; negative values indicate forgetting. As summarised in Table 3, HebbGate exhibits moderate forgetting ($-4$ to $-6$ %) on CIFAR-100 and ImageNet-100. This is consistent with strict parameter isolation: subnetworks for earlier tasks are not revisited by later gradients, so forgetting is bounded but non-zero due to shared classifier and partial overlap in early convolutional stages.

**Forward transfer, capacity allocation, and the role of $\kappa$-decay.** We quantify how much new tasks benefit from previous ones using a forward-transfer proxy $F$ (formal definition in App. A.3). Intuitively, $F$ measures the average gain in end-of-task Class-IL accuracy when using $\kappa$-decay instead of a static sparsity budget, under otherwise identical settings. The values in Table 3 show consistently positive forward transfer on CIFAR-100 and ImageNet-100 ($F \approx 7.3$–6.7 points), complementing the moderate forgetting indicated by BWT. Figure 3 visualises the end-of-task accuracy $\tilde{\mathcal{A}}(t, t)$ for each task on CIFAR-100 (ResNet-18). After the first task, $\kappa$-decay yields higher end-of-task accuracy for almost every task, and the gap tends to increase for later tasks despite their lower effective sparsity. This supports the intended interpretation of the $\kappa$-schedule: early tasks explore with larger masks, warming up the feature space, while later tasks start from more structured rep-

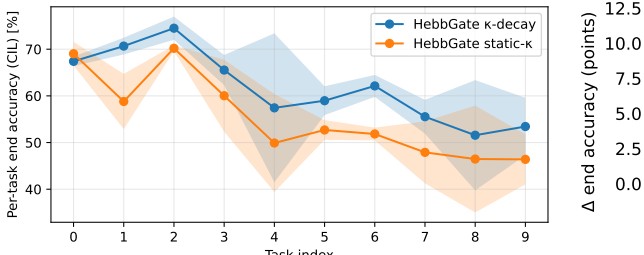 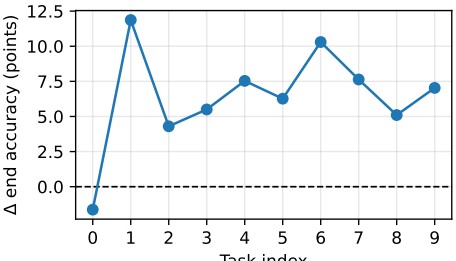

Figure 3: CIFAR-100, Class-IL (ResNet-18). Per-task end-of-task accuracy (mean $\pm$ s.d.) for HebbGate with $\kappa$-decay vs. static-$\kappa$. Later tasks achieve higher accuracy under $\kappa$-decay despite starting with lower effective sparsity, indicating positive forward transfer.

resentations and still benefit even under tighter capacity constraints. This pattern is mirrored in the learned mask-overlap matrices (App. Figs. 4) whose $(i, j)$ entry is the fraction of channels simultaneously active in tasks $i$ and $j$. With a fixed sparsity budget, early tasks on CIFAR-100 form dense overlap blocks in the first residual stage, indicating that they monopolise shallow-layer capacity. Under $\kappa$-decay and usage-aware gating, overlap is more evenly distributed across tasks, and the average pairwise overlap remains low. Taken together, these results show that the high initial $\kappa$ enables exploration, the decay phase prevents early-task saturation, and the resulting capacity allocation yields moderate forgetting (BWT) with positive forward transfer $F$.

**Additional ablations.** Table 4 summarises the impact of $\kappa$-decay versus static-$\kappa$ on CIFAR benchmarks. Across both Task-IL and Class-IL, $\kappa$-decay consistently matches or outperforms the static baseline. On CIFAR-10, Class-IL accuracy jumps $> 20\%$ when enabling $\kappa$-decay at the same final sparsity, and on CIFAR-100 from $51.6\%$ to $58.8\%$. Similar trends hold for AlexNet in App. A.6. Beyond the sparsity schedule, we ablate components of the Hebbian rule in App. A.6. Removing the reward signal (purely unsupervised update) still yields strong Task-IL performance on CIFAR-10 but reduces CIFAR-100 CIL to $\approx 53\%$. The margin-based reward recovers 10–22 % and reduces sensitivity to the choice of usage scaling. Varying the gate-update interval and evaluating on Permuted-MNIST further show HebbGate's robustness to these hyperparameters.

Table 4: Ablation on Split CIFAR-10 and CIFAR-100 (ResNet-18) under TIL and CIL settings. $\kappa$-decay denotes the warm-up schedule used in the main experiments; static-$\kappa$ uses the same final sparsity without warm-up.

| | CIFAR-10 | | | | | CIFAR-100 | | | |
|---|---|---|---|---|---|---|---|---|---|
| | **TIL Acc.** | | **CIL Acc.** | | | **TIL Acc.** | | **CIL Acc.** | |
| $\kappa$ | $\kappa$-decay | static-$\kappa$ | $\kappa$-decay | static-$\kappa$ | $\kappa$ | $\kappa$-decay | static-$\kappa$ | $\kappa$-decay | static-$\kappa$ |
| 0.20 | **94.7±0.2** | 89.6±1.2 | **92.3±1.1** | 69.5±4.2 | 0.10 | **71.1±0.7** | 58.0±0.5 | **58.8±0.9** | 51.6±1.3 |
| 0.25 | 90.1±3.4 | 81.7±2.5 | 76.5±5.7 | 70.5±4.1 | 0.125 | 65.7±1.1 | 67.5±0.7 | 56.5±1.8 | 51.1±1.3 |

## 6 CONCLUDING REMARKS

This paper proposes HebbGate, an efficient and scalable continual-learning method that uses a fully local, reward-modulated Hebbian rule that avoids the computational and memory overhead of gradient-based masking methods. By allocating compact sub-networks without relying on exemplars or auxiliary networks, HebbGate achieves strong accuracy with significantly reduced training time and minimal memory growth, making it a practical choice for exemplar-free, single-head continual learning. HebbGate still assumes (i) known task boundaries and a bounded task count $T$ to schedule its $\kappa$-decay, (ii) semantically coherent classes within each task, and (iii) vision-only streams up to $T{=}10$. Future work will (a) extend the greedy $\kappa_t = \kappa_0(1 - t/\tau)$ schedule to fully *unknown-T* streams, (b) compress the gate bank for very long task sequences, and (c) generalise the local Hebbian gating principle to domain-incremental and NLP benchmarks.

REPRODUCIBILITY STATEMENT

For all comparative results we run each experiment five times with different random seeds and report the mean ± standard deviation. Full training and evaluation details − including hardware, optimizer choices, searched ranges and selected hyperparameters (Table 5), backbone definitions (Table 6), dataset splits, and protocols are provided in Section A.3 and Appendices A.2–A.4. The Class-IL scoring and calibration procedure is described with pseudocode in Appendix A.4. We will release the source code (including scripts to regenerate tables/CSVs) upon acceptance to enable exact reproduction of our results.

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

# A    TECHNICAL APPENDICES AND SUPPLEMENTARY MATERIAL

In these appendices we provide additional information, experimental results, and analyses that complement the main paper.

**Scope of the Appendices.**    In Appendix A.1, we compile the default notation used throughout the work, covering the incremental setup, gating variables, sparsity schedule, and update coefficients. In Appendix A.2, we report complete reproducibility details, including hardware, optimizer choices, hyperparameter ranges; then enumerates training configurations (backbones, layer shapes). In Appendix A.3, we specify datasets and evaluation protocols. In Appendix A.4, we detail Class-IL inference via fused/parallel gating and $z$-normalization of task scores, and we provide complete pseudocode for the calibration and selection procedure. In Appendix A.5, we report extended Task-IL and Class-IL results for both AlexNet and ResNet-18. In Appendix A.6, we present extensive ablations on $\kappa$-decay versus static sparsity, gate-update rate, and rule components, along with overlap analyses that quantify subnetwork reuse. In Appendix A.7, we explore variations of the Hebbian update itself, contrasting reward signals, capacity-aware scaling rules, and hard/soft masking alternatives, backed by comparative results. In Appendix A.8, we analyze latency and peak-memory trade-offs, including the speedup from the fused parallel pass versus sequential evaluation and the parameter-footprint implications relative to baselines. Finally, in Appendix A.9, we discuss limitations and future directions.

## A.1    DEFAULT NOTATION

For convenience, we summarize here the notation used in the paper.

**Incremental learning setup.**

- $\mathcal{M}_t$: the incremental model at task $t$ (see Sec. 4).
- $T$: total number of tasks in the sequence.
- $\mathbb{C}_t$: set of classes associated with task $t$; $|\mathbb{C}_t|$ is its cardinality.
- $\mathbb{D}$: incremental dataset; at task $t$ it provides samples $\mathbb{X}_t$ and labels $\mathbb{Y}_t$.
- $f(\cdot; \boldsymbol{\theta}_t)$: backbone feature extractor at task $t$ with parameters $\boldsymbol{\theta}_t$.
- $n$: feature dimensionality, i.e., $f(\boldsymbol{x}; \boldsymbol{\theta}_t) \in \mathbb{R}^n$.
- $\boldsymbol{W}_t \in \mathbb{R}^{n \times \sum_{j=1}^{t} |\mathbb{C}_j|}$: linear classifier at task $t$ (single-head Class-IL).

**Prediction and losses.**

- $\boldsymbol{z} \in \mathbb{R}^{|\mathbb{C}_{\leq t}|}$: logit vector over classes $\mathbb{C}_{\leq t} := \bigcup_{j=1}^{t} \mathbb{C}_j$; element $c$ is $z_c$.
- $\hat{y} = \arg\max_{c \in \mathbb{C}_{\leq t}} z_c$: predicted class.
- $\ell(\hat{y}, y)$: task-agnostic classification loss used for training in Sec. 4.

**HebbGate: utilization-aware gating (overview in Fig. 1).**

- $k \in \{1, \ldots, K\}$: channel (neuron/filter) index in a layer; $K$ channels total.
- $\boldsymbol{a} \in \mathbb{R}^K$: channel activation vector (per layer, per input or mini-batch).
- $g_{k,t} \in \mathbb{R}$: scalar gate stored for channel $k$ and task $t$ (gate bank).
- $\boldsymbol{g}_t \in \mathbb{R}^K$: gate vector for task $t$, collecting $\{g_{k,t}\}_{k=1}^{K}$.
- $\kappa \in (0, 1]$: target sparsity (fraction of active channels); $\kappa_t$ is the scheduled value at time/epoch $t$.
- $\boldsymbol{m}_t \in \{0, 1\}^K$: binary mask for task $t$ selecting the top-$\kappa_t K$ channels by gate score.
- $\boldsymbol{a} \odot \boldsymbol{m}_t$: Hadamard product; masked activations forwarded through the backbone.

**Local three-factor learning signals.**

- $E_k \geq 0$: activation "energy" statistic for channel $k$ (batch-level).
- $r \in \mathbb{R}$: signed margin reward (positive when predictions are confident/correct, negative otherwise).
- $u_k \in [0, 1]$: running usage estimate for channel $k$; $\phi(u_k)$: usage-aware scaling (power-law penalty).

**HebbGate update and schedules (see Sec. 4).**

- $\eta_g > 0$: gate update step size; $\lambda_g \geq 0$: gate decay/forgetting coefficient.
- **Gate update (per channel):** $g_{k,t} \leftarrow (1 - \lambda_g) g_{k,t} + \eta_g E_k r \phi(u_k)$.
- $\kappa$**-decay:** schedule controlling sparsity $\kappa_t$ (exploration $\rightarrow$ consolidation).

**Indices & sets (ICLR style).**

- $t \in \{1, \ldots, T\}$: task index; $c \in \mathbb{C}_t$: class index within task $t$.
- $\mathbb{B}$: mini-batch; $\mathbb{R}$: set of real numbers.

## A.2 REPRODUCIBILITY DETAILS

Appendix A.2 provides extended reproducibility details that complement the experimental results in Section *Experiments*. All models were trained on an NVIDIA RTX 4080 GPU with an Intel i9–13900K CPU. Code will be released upon acceptance[5].

| Operation - Hyperparmeters | Searched range | Chosen |
|---|---|---|
| Batch Size | [64, 128] | 128 |
| Optimizer | Adam/ SGD | Adam |
| BP Weights LR | [0.01-0.00005] | 0.0001 |
| Hebbian Gate LR | [0.0001-0.0003] | 0.0001 |
| ResNet Normalization | BN, GN, IN | IN, Per-task BN |
| AlexNet Normalization | BN, GN, IN | GN |
| $\kappa$ (CIFAR-10) | [0.18, 0.2, 0.22, 0.25] | 0.2 |
| $\kappa$ (CIFAR-100) | [0.1, 0.125, 0.15, 0.2] | 0.1 |
| $\kappa$ (ImageNet-100) | [0.1, 0.125, 0.15, 0.2] | 0.1 |
| $\kappa$ (Tiny-ImageNet-200) | [0.1, 0.125, 0.15] | 0.1 |
| CIFAR-10 Epochs | P: 15-30; W: 10:30 | P: 15; W: 15 |
| CIFAR-100 Epochs | P: 15-60; W: 10:40 | P: 55; W: 25 |
| ImageNet-100 Epochs | P: 15-60; W: 10:40 | P: 55; W: 25 |
| Tiny-ImageNet-200 Epochs | P: 15-60; W: 10:40 | P: 55; W: 25 |
| Gate Update Rate | [5, 10, 20, 30] | [10-20] |

Table 5: Hyperparameter choices for the main configurations in the ablation studies. 'P' and 'W' Epochs denote the number of epochs trained under Parallel and Weight phases, respectively. BN, GN, IN refer to BatchNorm, GroupNorm, and InstanceNorm normalizations respectively.

### A.2.1 TRAINING CONFIGURATIONS

**Backbones.** Table 6 lists the architectures used in our experiments. ResNet-18 is the main backbone for CIFAR-100, Tiny-ImageNet-200, and ImageNet-100. For ablations and comparison with earlier parameter-isolation work, we additionally use a 3-conv AlexNet and a two-layer MLP-2000. Where needed to reproduce published baselines (e.g., HAT, EWC, LwF), we also report results on a downsized AlexNet-1024 configuration following community practice.

---

[5]A complete CSV with every hyper-parameter $\times$ accuracy ($\mu/\sigma$) is generated by the script make_ablation_table.py included in the code release.

All convolutional layers use either InstanceNorm or GroupNorm in the main HebbGate-IN configuration to avoid BatchNorm running-statistic leakage across tasks; fully-connected layers on AlexNet, two-layer MLP apply LayerNorm. The HebbGate-BN variant replaces these with task-specific BatchNorm statistics while keeping the rest of the architecture unchanged. Apart from gating and normalisation choice, the networks match standard community baselines.

| Backbone | Layers / channels | # Params |
|---|---|---|
| **AlexNet** | Conv $3\times3$ @ $64$ – GN
Conv $3\times3$ @ $128$ – GN
Conv $3\times3$ @ $256$ – GN
GAP $\to$ FC $256 \to C$ | 0.5M |
| **AlexNet-1024** | Conv $5\times5$ @ $64$ – GN, MaxPool $3\times3$ $s\!=\!2$
Conv $3\times3$ @ $128$ – GN, MaxPool $3\times3$ $s\!=\!2$
Conv $3\times3$ @ $256$ – GN, MaxPool $3\times3$ $s\!=\!2$
FC $256\cdot3\cdot3 \to 1024$ – LN
FC $1024 \to 2048$ – LN
FC $2048 \to C$ | 6.5 M |
| **Gated ResNet-18** | Stem $3\times3$ @ $64$ – IN (or Per-task BN)
4 residual stages $\{64, 128, 256, 512\}$ @ 2 blocks
GAP $\to$ FC $512 \to C$ | 11.2M |
| **MLP-2000** | FC $3\cdot32^2 \to 2000$ – LN $\times2$
FC $2000 \to C$ | 8.3M |

Table 6: Architectures used in all experiments ($C$ = number of classes for the dataset). "GAP" denotes global average pooling.

**Optimisation.** Following the training loop and phase schedule described in the main text, each task is trained in two phases (Alg. 1):

- *Parallel phase*: weights are updated *every* mini-batch in parallel with gates, which are updated at a slower rate (Table 5).
- *Weight-only phase*: gates are frozen and weights are fine-tuned.

The number of epochs in each phase depends on the dataset and backbone and is summarised in Table 5. We use Adam with learning rate $10^{-4}$ for the backbone weights. Gate learning uses a Hebbian learning rate $\eta_g \in \{1, 3\} \times 10^{-4}$ and is applied every `gate_update_rate` $\in \{10, 20\}$ mini-batches.

The active fraction starts at $\kappa_{\text{start}} \in [0.5, 0.65]$ and is linearly annealed to $\kappa_{\min} \approx 1/T$ during the parallel phase of the current task only. This schedule encourages early exploration and later consolidation, and consistently improves Class-IL accuracy over a fixed sparsity budget across both backbones (see Table 4).

**Evaluation protocol.** We report Task-IL (single oracle head) and 10-task cold-start Class-IL (single shared head) accuracies at the end of each task. Unless otherwise stated, all reported numbers are averaged over five random seeds ($\mu \pm \sigma$). Cold-start or zero-shot accuracy refers to the test accuracy obtained on a previously unseen task *immediately* after the model has finished training on the earlier tasks.

**Continual-learning metrics.** Let $\mathcal{A}(t, i)$ denote the Top-1 accuracy on the test set of task $i$ after completing training on task $t$. Following Chaudhry et al. (2019); Magistri et al. (2024); Goswami et al. (2023); Gomez-Villa et al. (2024), we define:

$$A_{\text{last}} = \frac{1}{T} \sum_{i=1}^{T} \mathcal{A}(T, i), \qquad A_{\text{inc}} = \frac{1}{T} \sum_{t=1}^{T} \left( \frac{1}{t} \sum_{i=1}^{t} \mathcal{A}(t, i) \right),$$

and use $A_{\text{last}}$ and $A_{\text{inc}}$ as our main accuracy metrics. We also report backward transfer (BWT) and forgetting:

$$\mathrm{BWT} = \frac{1}{T-1} \sum_{i=1}^{T-1} \left( \mathcal{A}(T,i) - \mathcal{A}(i,i) \right),$$

with forgetting defined analogously using negative improvements.

To quantify forward transfer from earlier to later tasks, we use a forward-transfer proxy $F$ when comparing two configurations that are identical except for the sparsity schedule (e.g. $\kappa$-decay vs. static-$\kappa$). Let $\tilde{\mathcal{A}}^{\mathrm{decay}}(t,t)$ and $\tilde{\mathcal{A}}^{\mathrm{static}}(t,t)$ denote the *end-of-task* Class-IL accuracy on task $t$ for the decayed and static variants, respectively. We define

$$F = \frac{1}{T-1} \sum_{t=2}^{T} \left( \tilde{\mathcal{A}}^{\mathrm{decay}}(t,t) - \tilde{\mathcal{A}}^{\mathrm{static}}(t,t) \right),$$

so that $F > 0$ indicates that, on average, later tasks learn better under $\kappa$-decay than under a fixed sparsity budget.

## A.3 DATASETS AND EVALUATION PROTOCOLS

Continual–learning behaviour is highly sensitive to the choice of benchmark. To provide a balanced evaluation across semantic complexity, input resolution, and task heterogeneity, we use three Class-IL benchmarks and two secondary benchmarks used primarily for ablations and sanity checks. Table 7 summarises their key properties.

| Dataset | Resolution | Stream length | Classes / task | Total classes | Images / task (train / test) | IL setting |
|---|---|---|---|---|---|---|
| CIFAR-100 Krizhevsky (2009) | $3 \times 32 \times 32$ | **10** | 10 | 100 | $5\,000/1\,000$ | C |
| Tiny-ImageNet-200 Fei-Fei et al. (2015) | $3 \times 64 \times 64$ | **10** | 20 | 200 | $10\,000/2\,000$ | C |
| ImageNet-100 Deng et al. (2009) | $3 \times 224 \times 224$ | **10** | 10 | 100 | $\approx 13\,000/500$ | C |
| Split-CIFAR-10 Krizhevsky (2009) | $3 \times 32 \times 32$ | **5** | 2 | 10 | $10\,000/2\,000$ | T / C |
| Permuted-MNIST LeCun et al. (1998) | $1 \times 28 \times 28$ | **10** | 10 | 10 | $6\,000/1\,000$ | T |

Table 7: Benchmarks used in this work. The *stream length* is the number of tasks, *classes / task* the number of classes presented per task, and *IL setting* indicates whether we report results in the Task-IL (T) or Class-IL (C) protocol. The first three datasets form our main Class-IL evaluation; the remaining ones are used for additional ablations.

**CIFAR-100 (main).** We use the standard 10-task split with 10 new classes per task (100 total), evaluated only in Class-IL. Images are $32 \times 32$ with standard augmentation (random crop with 4-pixel padding, horizontal flip) and channel-wise mean/std normalisation. This benchmark provides a medium-complexity, long sequence for studying capacity allocation and forgetting.

**Tiny-ImageNet-200 (main).** Tiny-ImageNet-200 has 200 classes of $64 \times 64$ images. We follow recent CL work and form 10 tasks of 20 classes each, evaluated in Class-IL. Training uses random resized crops and horizontal flips; testing uses centre crops. Compared to CIFAR-100, it increases both resolution and class granularity.

**ImageNet-100 (main).** ImageNet-100 is a 100-class subset of ImageNet with $224 \times 224$ images, split into 10 tasks of 10 classes each and evaluated only in Class-IL. We use the standard ImageNet augmentation pipeline. This benchmark tests scalability to high-resolution, large-distribution data.

**Permuted-MNIST (secondary).** Each of the 10 tasks applies a fixed random pixel permutation to MNIST images ($28 \times 28$), preserving labels but destroying spatial structure. We use only the Task-IL setting. This benchmark mainly probes global weight stability rather than feature reuse.

**Split-CIFAR-10 (secondary).** Split-CIFAR-10 forms 5 tasks with 2 classes each (10 total). We use the same augmentations as CIFAR-100 (random crop + flip) and report both Task-IL and Class-IL in ablations. This benchmark tests feature reuse under mild semantic shifts.

CIFAR-100, Tiny-ImageNet-200, and ImageNet-100 cover a spectrum from low-resolution to realistic high-resolution data and form our main Class-IL evaluation. Permuted-MNIST, and Split-CIFAR-10 complement these benchmarks by probing gradient obstruction, feature reuse, and sensitivity to hyperparameter choices. HebbGate performs consistently across both main and secondary benchmarks, highlighting the generality of the proposed local gating mechanism.

## A.4 DETAILS FOR CLASS-IL INFERENCE AND $z$-NORMALIZATION

### A.4.1 CALIBRATION SPLIT AND STATISTICS

We estimate per–task moments $(\mu_t, \sigma_t)$ of the confidence scores $s_t(x) = \max_c \mathrm{softmax}(f(x;t))_c$ on a small calibration set $\mathbb{D}_{\mathrm{cal}} = \{x_i\}_{i=1}^{N_{\mathrm{cal}}}$ (disjoint from the training stream; class labels are not required). For each task $t \in \{1, \ldots, T\}$,

$$\mu_t \;=\; \frac{1}{N_{\mathrm{cal}}} \sum_{i=1}^{N_{\mathrm{cal}}} s_t(x_i), \qquad \sigma_t \;=\; \sqrt{\frac{1}{N_{\mathrm{cal}}} \sum_i s_t(x_i)^2 - \mu_t^2}. \tag{13}$$

In practice we compute equation 13 on mini-batches and maintain running sums $\sum s_t$ and $\sum s_t^2$. We floor the standard deviation as $\sigma_t \leftarrow \max(\sigma_t, \sigma_{\min})$ with $\sigma_{\min} \in [10^{-3}, 10^{-2}]$ to avoid division by very small values when a task's score distribution is extremely concentrated. $\mathbb{N}_{\mathrm{cal}} = 5\text{--}20$ batches for $B = 64 - 128$ is sufficient in our experiments. Larger $\mathbb{N}_{\mathrm{cal}}$ yields diminishing returns.

### A.4.2 INFERENCE WITH $z$-NORMALIZED SCORES

Given a test input $x$, define the normalized score

$$\tilde{s}_t(x) \;=\; \frac{s_t(x) - \mu_t}{\sigma_t + \varepsilon}, \qquad \varepsilon > 0 \text{ (e.g., } 10^{-8}). \tag{14}$$

We select the task by $\hat{t}(x) = \arg\max_t \tilde{s}_t(x)$ and predict with $f(x; \hat{t})$. For a mini-batch $\{x_i\}_{i=1}^B$ we compute $\tilde{s}_t(x_i)$ for all $t$ and either:

- *Per-sample selection* (recommended): $\hat{t}(x_i) = \arg\max_t \tilde{s}_t(x_i)$, then use $f(x_i; \hat{t}(x_i))$.

- *Batch selection* (used in fused evaluation): average $\bar{s}_t = \frac{1}{B} \sum_i \tilde{s}_t(x_i)$ and take $\hat{t} = \arg\max_t \bar{s}_t$, then use $f(x_i; \hat{t})$ for all $i$ in the batch.

Both modes use the same calibrated scores; the former generally yields higher accuracy but the latter aligns exactly with the single fused $T$-way forward.

### A.4.3 PSEUDOCODE

---

**Algorithm 2** Class-IL with $z$-normalized task scores (no learned calibrators)

---

1: **Input:** model $f(\cdot;\cdot)$, tasks $\{1,\dots,T\}$, calibration set $X \subset \mathbb{D}_{\text{cal}}$, test batch $\{x_i\}_{i=1}^B$
2: **Calibration:** set `eval()`, initialize $\mu_t \leftarrow 0$, $\nu_t \leftarrow 0$, $n \leftarrow 0$ for all $t$
3: **for** minibatch $X \subset \mathbb{D}_{\text{cal}}$ **do**
4:    **for** $t = 1$ **to** $T$ **do**
5:       compute $s_t(X) = \max_c \text{softmax}(f(X;t))_c$ elementwise
6:       $\mu_t \mathrel{+}= \sum s_t(X), \quad \nu_t \mathrel{+}= \sum s_t(X)^2, \quad n \mathrel{+}= |X|$
7:    **end for**
8: **end for**
9: $\mu_t \leftarrow \mu_t/n, \quad \sigma_t \leftarrow \sqrt{\nu_t/n - \mu_t^2}, \quad \sigma_t \leftarrow \max(\sigma_t, \sigma_{\min})$
10: **Inference:** for test batch $\{x_i\}_{i=1}^B$
11: **for** $t = 1$ **to** $T$ **do**
12:    compute $s_t(x_i)$ for all $i$
13: **end for**
14: $\tilde{s}_t(x_i) \leftarrow \dfrac{s_t(x_i) - \mu_t}{\sigma_t + \varepsilon}$         (vectorized over $i$)
15: **Per-sample:** $\hat{t}(x_i) = \arg\max_t \tilde{s}_t(x_i); \quad \hat{y}(x_i) = \arg\max_c f(x_i;\hat{t}(x_i))_c$
16: **(or) Batch:** $\bar{s}_t = \frac{1}{B}\sum_i \tilde{s}_t(x_i); \quad \hat{t} = \arg\max_t \bar{s}_t; \quad \hat{y}(x_i) = \arg\max_c f(x_i;\hat{t})_c$

---

## A.5 EXTENDED COMPARISON WITH RELATED WORK

This is an extended comparison with related work with Task-IL settings on both AlexNet and ResNet backbones and the PERMUTED-MNIST ;SPLIT-CIFAR-10 ;SPLIT-CIFAR-100 Goodfellow et al. (2013); Krizhevsky (2009) datasets. Table 8 reports mean±std over five seeds under the **10-step cold-start** protocol.

On **CIFAR-10 (Class-IL)**, HebbGate achieves new SOTA on both backbones (84.5±5.8% on ResNet-18), markedly outperforming the strongest distillation baseline (LwF, 63.5±1.5%). On **CIFAR-100 (Class-IL)**, HebbGate achieves the highest overall accuracy using the AlexNet backbone (60.6±0.5%), surpassing GPM and HAT by 5 %. In **CIFAR-100 (TIL)**, despite not using MH at test time, HebbGate remains competitive with SPG and PathNet. In terms of efficiency and memory, HebbGate stores only per-channel gate scalars; thus, its parameter footprint matches the base network. In contrast, HAT's float masks inflate memory roughly ×10 and EWC roughly ×2 after $T{=}10$ tasks (Appendix A.8, Table 14). At inference, a single fused forward executes all gates in 9.6× faster than a sequential loop and $\geq$ 10× faster than other methods measured under our setup. The strongest regularization/distillation baselines require heavier training (auxiliary penalties/teachers and longer schedules), leading to substantially higher compute cost. Specifically, EFC requires > 21 training time per epoch under our setup despite similar parameter footprints.

## A.6 EXTENSIVE ABLATION

In this section, we evaluate the effect of our architectural choices, hyperparameter settings, and individual components. Detailed results and full comparative tables are provided in Appendix. We observe that the $\kappa$–decay warm-up yields substantial gains in Task-IL performance on Split-CIFAR-100: an average accuracy improvement of +8% with the AlexNet backbone and +6% with gated ResNet-18. Although the warm-up temporarily grants each task access to a larger fraction of units, our usage-aware scaling still bounds inter-task overlap and mitigates forgetting.

To quantify sub-network reuse, we compute the neuron-mask overlap between every pair of tasks. Figure 5 displays the resulting overlap matrix, where entry $(i, j)$ is the fraction of neurons selected by task $i$ that are also used by task $j$. Across all layers and backbones, overlaps remain below 11% in nearly all cases, confirming that HEBBGATE enforces tight parameter isolation while permitting limited, targeted sharing when beneficial.

Table 8: Average accuracy (%, ↑) on: (a) Task-IL (TIL) setting on CIFAR-100; **10-step cold-start Class-IL** protocol on (b) CIFAR-100; and (c) CIFAR-10. Values are the mean ± standard deviation over five seeds. * denotes sigma not reported. (MH) denotes *multi-head* classifier inference.

(a) CIFAR-100 — TIL

| | Method | TIL Acc. | MH |
|---|---|---|---|
| AlexNet | HAT | $62.8 \pm 0.7$ | ✓ |
| | PathNet | $\mathbf{69.1 \pm 0.5}$ | ✓ |
| | SPG | $\underline{67.7 \pm 0.3}$ | ✓ |
| | EWC | $61.6 \pm 0.9$ | ✓ |
| | **HebbGate** (ours) | $67.3 \pm 1.2$ | – |

(b) CIFAR-100 — CIL

| | Method | CIL Acc. |
|---|---|---|
| AlexNet | HAT Serra et al. (2018) | $52.7 \pm 1.0$ |
| | GPM Saha et al. (2021) | $\underline{55.5 \pm 1.7}$ |
| | **HebbGate** (ours) | $\mathbf{60.6 \pm 0.5}$ |

(c) CIFAR-10 — CIL

| | Method | CIL Acc. | MH |
|---|---|---|---|
| ResNet | PASS | $46.5$ * | – |
| | EWC | $60.0 \pm 2.3$ | – |
| | LwF | $\underline{63.5 \pm 1.5}$ | ✓ |
| | **HebbGate** (ours) | $\mathbf{84.5 \pm 5.8}$ | – |
| AlexNet | HAT | $\underline{79.7 \pm 1.3}$ | ✓ |
| | **HebbGate** (ours) | $\mathbf{82.4 \pm 6.8}$ | – |

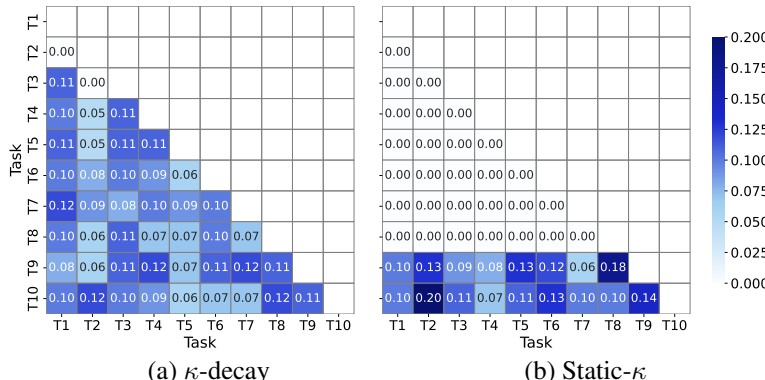

(a) $\kappa$-decay      (b) Static-$\kappa$

Figure 4: Inter-task overlap on CIFAR-100 (AlexNet-CNN).

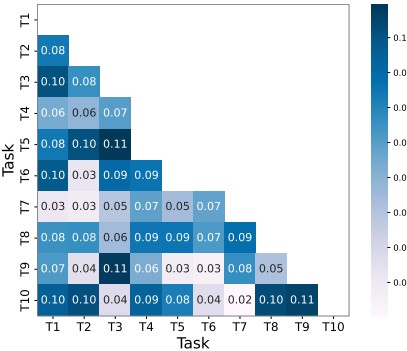

Figure 5: Pairwise neuron mask overlap between 10 tasks of CIFAR-100 with static-$\kappa$. Each cell $(i, j)$ represents the proportion of neurons reused from task $i$ when learning task $j$. Values above the diagonal are masked for clarity. Tasks are indexed $T_1$ to $T_{10}$. The matrix highlights how overlap remains limited across most task pairs, demonstrating the system's capacity-aware gating and selective reuse.

| Dataset | Architecture | $\kappa$ | Gate Update Rate | TIL Accuracy | | CIL Accuracy | |
|---|---|---|---|---|---|---|---|
| | | | | $\kappa$-Decay | Static-$\kappa$ | $\kappa$-Decay | Static-$\kappa$ |
| CIFAR-10 | ResNet | 0.2 | 10 | 90.1 ± 5.1 | 77.7 ± 1.6 | 64.8 ± 11.3 | 63.0 ± 6.1 |
| | | | 20 | **91.0 ± 0.6** | 83.6 ± 1.9 | **84.5 ± 5.8** | 56.4 ± 8.6 |
| | | 0.25 | 10 | 86.3 ± 8.8 | 74.3 ± 11.6 | 66.2 ± 12.8 | 75.4 ± 3.3 |
| | | | 20 | 89.4 ± 4.0 | 80.4 ± 2.7 | 77.3 ± 6.4 | 58.5 ± 4.2 |
| | AlexNet | 0.2 | 10 | **91.2 ± 3.0** | 82.5 ± 2.7 | **89.4 ± 3.7** | 76.7 ± 7.2 |
| | | | 20 | 90.2 ± 1.3 | 84.4 ± 2.7 | 82.4 ± 6.8 | 69.7 ± 7.2 |
| | | 0.25 | 10 | 86.8 ± 1.8 | 82.8 ± 5.8 | 68.5 ± 13.4 | 34.8 ± 24.6 |
| | | | 20 | 90.1 ± 2.4 | 84.1 ± 2.8 | 73.0 ± 9.8 | 54.1 ± 8.8 |
| CIFAR-100 | ResNet | 0.1 | 10 | 62.5 ± 4.3 | 47.9 ± 0.9 | 50.3 ± 4.7 | 32.3 ± 2.7 |
| | | | 20 | 62.25 ± 0.6 | 53.4 ± 0.7 | 54.5 ± 2.1 | 47.8 ± 1.1 |
| | | 0.125 | 10 | 63.9 ± 1.9 | 54.1 ± 0.6 | 51.9 ± 4.9 | 53.9 ± 1.9 |
| | | | 20 | **64.1 ± 1.3** | 54.6 ± 0.6 | **54.5 ± 2.3** | 47.1 ± 0.4 |
| | AlexNet | 0.1 | 10 | 63.5 ± 0.1 | 64.5 ± 0.2 | 56.0 ± 6.6 | 54.4 ± 0.2 |
| | | | 20 | **67.3 ± 1.2** | 65.2 ± 0.7 | 60.6 ± 0.5 | 59.2 ± 0.6 |
| | | 0.125 | 10 | 62.4 ± 1.1 | 65.3 ± 0.8 | 58.6 ± 4.8 | 57.3 ± 1.5 |
| | | | 20 | 66.2 ± 1.2 | 65.5 ± 0.5 | 60.2 ± 3.1 | **60.8 ± 2.7** |

Table 9: Ablation table with overall accuracy on the Task-Incremental learning and Class-Incremental learning settings of Split-CIFAR benchmarks, for the ResNet-18 and AlexNet architectures, $\kappa$ fractions, and gate update rates.

### A.6.1 SPLIT-CIFAR

**Effect of $\kappa$, gate–update rate, and warm-up on final accuracy**  Table 9 reports the *end-of-stream* accuracy (%; mean ± s.d. over three runs) for every combination of

1. backbone (ResNet-18 vs. AlexNet),

2. sparsity budget ($\kappa \in \{0.10, 0.125\}$ for CIFAR-100,  $\kappa \in \{0.20, 0.25\}$ for CIFAR-10),

3. gate-update interval ($\upsilon \in \{10, 20\}$ mini-batches),

4. *with* vs. *without* the proposed $\kappa$-decay warm-up,

5. evaluation protocol (Task-IL *vs.* Class-IL).

Warm-up is consistently beneficial − especially for Class-IL.  Across *all* 16 configurations the warm-up variant outperforms (or matches within the confidence band) the static-$\kappa$ counterpart. The average gain is +3.1 % in Task-IL and +6.7 % in Class-IL. The impact of $\kappa$-decay when the capacity is tight is stronger over Task-IL settings: for ResNet-18 on CIFAR-100, top-$k$ decay lifts Class-IL across different sets of hyperparameters, a mere *+1.1 %* jump.

Longer gate-update intervals help large backbones.  Moving from $\upsilon = 10$ to $\upsilon = 20$ updates per epoch gives a small but systematic boost on ResNet-18 (average +1.8 % Task-IL, +2.3 % Class-IL), while AlexNet - which contains 4 × fewer layers − shows no significant change. A plausible explanation is that deeper networks accumulate more stochastic variance in the Hebbian signals; fewer, larger updates let the reward signal integrate over multiple weight steps and stabilise the gate trajectories.

Sparsity budget trades plasticity for stability.  Increasing $\kappa$ from 0.10 to 0.125 on CIFAR-100 (or $0.20 \to 0.25$ on CIFAR-10) yields +3.5 % Task-IL on ResNet, but only +0.8 % on AlexNet. The deeper models' tighter bottleneck is at a disadvantage when the allowed shared capacity is tighter.

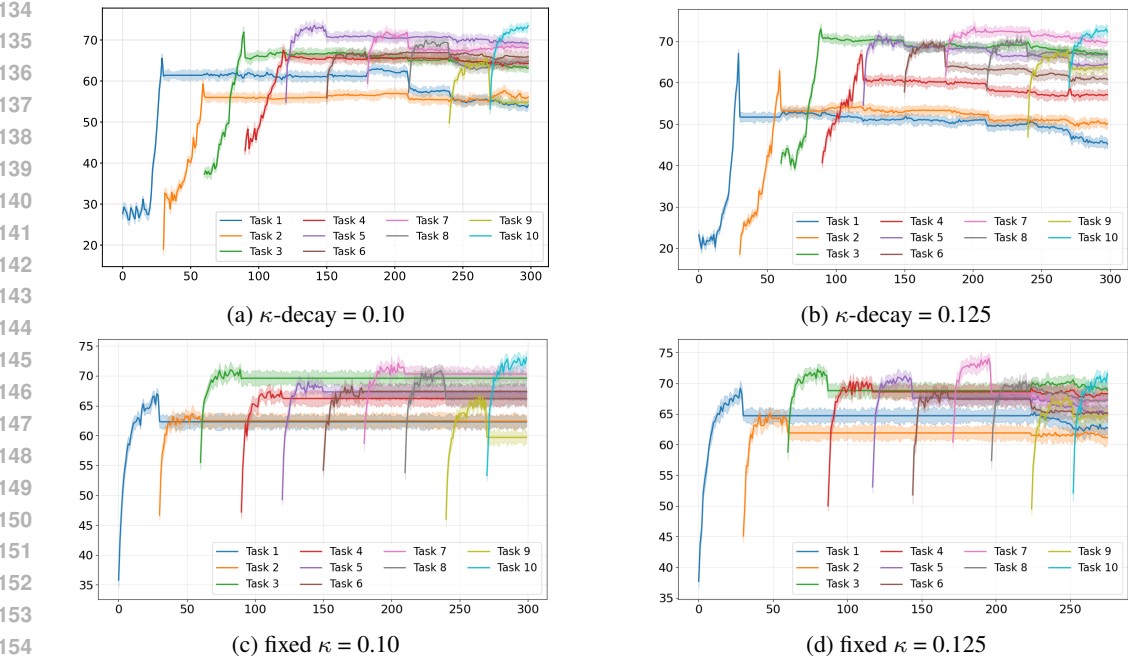

Figure 6: Plots with per-task accuracies on AlexNet for Split-CIFAR-100 (Task-IL Setting) and the effect of $\kappa$-decay warm-up. Each panel plots mean $\pm$ s.d. of 3 runs for all 10 tasks (Task 1–10, colour-coded). Top row: our linear $\kappa$-decay warm-up; bottom row: the same runs with a fixed sparsity budget. Left column uses $\kappa = 0.10$, right uses $\kappa = 0.125$. Common hyper-parameters: batch 128, gate-update interval 20, $\texttt{gating\_LR} = 10^{-4}$, BP learning-rate $= 10^{-4}$, margin reward, POWER usage scaling, hard masks.

Instead the shallower AlexNet, whose early convolutional stage is the bottleneck is not affected at the same level. Notably, for *static-*$\kappa$ runs the larger budget translate into proportional Class-IL gains.

**Take-away.** For CIFAR-10 the top performer is AlexNet, $\kappa$-decay $+ \kappa = 0.20$, $\upsilon = 10$ ($91.2 \pm 3.0$ Task-IL, $89.4 \pm 3.7$ Class-IL). On CIFAR-100 the combination AlexNet, $\kappa = 0.125$, $\upsilon = 20$} delivers the highest overall scores (Static $\kappa$: $65.5 \pm 0.5$), ($\kappa$-Decay: $66.2 \pm 1.2$) Task-IL, (Static $\kappa$: $60.8 \pm 2.7$), ($\kappa$-Decay: $60.2 \pm 3.1$) Class-IL. These settings are adopted as the default hyper-parameters in the main paper. The ablation confirms that HEBBGATE retains high performance over a broad hyper-parameter range and that its gains do not hinge on a fragile choice of $\kappa$ or update rate.

**Task-Incremental Per-task accuracies.** We ablate the proposed $\kappa$-decay schedule on SPLIT-CIFAR-100 with the *gated* AlexNet. Each curve reports *mean $\pm$ s.d.*. Figure 6 contrasts two capacity budgets ($\kappa = 0.10$ on the left, $\kappa = 0.125$ on the right) with and without the linear warm-up ($\kappa_{\text{start}} = 0.5 \rightarrow \kappa_{\min}$).

With $\kappa$-decay (panels 6a–6b) the largest drop never exceeds 4–5 %. Eliminating the warm-up (panels 6c–6d) along with low $\kappa$, leads to minimal forgetting, however, it doesn't reach the peak per-task accuracy of the other combinations.

After Task 3, the blue/orange/green traces in Fig. 6a remain essentially flat, confirming that the *usage-aware scaling* combined with a shrinking $\kappa$ protects earlier subnetworks. With a fixed mask (Fig. 6c) those same traces drift downward by 3–5 %.

**Gate-overlap analysis.** Figure 5 reports the layer-averaged *mask-overlap matrices* for the same two configurations evaluated in Table 9 and Figure 6a-6d. With $\kappa$-decay (panel 4a) pairwise reuse remains tightly clustered around the target sparsity: most entries stay below $0.12$ and decay monotonically away from the diagonal, indicating that successive tasks recruit *fresh* channels while re-using only a minority of highly generic ones. This balanced allocation explains why the peak forgetting in Figures 6a-6b, never exceeds $4-5$ % and why the Class-IL accuracy in Table 9 rises despite the

Table 10: Forgetting vs. reuse (AlexNet, CIFAR-100).

| Setting | Avg. overlap | Avg. forgetting |
|---|---|---|
| $\kappa$-decay | 0.08 | 4.3 % |
| static-$\kappa$ | 0.18 | 12.4 % |

| Architecture | Reward | Type | $\kappa$ | No Reward | | Reward (Margin) | |
|---|---|---|---|---|---|---|---|
| | | | | TIL Acc. | CIL Acc. | TIL Acc. | CIL Acc. |
| ResNet | exp | hard | 0.18 | 77.02 ± 5.7 | 40.64 ± 9.2 | 98.63 ± 3.6 | 74.35 ± 7.8 |
| | | | 0.22 | 69.19 ± 1.8 | 39.03 ± 14.2 | 98.60 ± 1.3 | 72.34 ± 4.9 |
| | | soft | 0.18 | 84.28 ± 2.3 | 53.09 ± 9.1 | 94.79 ± 1.9 | 70.87 ± 5.2 |
| | | | 0.22 | 69.33 ± 1.9 | 45.88 ± 4.0 | 95.96 ± 1.5 | 59.67 ± 3.3 |
| | linear | hard | 0.18 | 88.63 ± 4.2 | 40.73 ± 23.4 | 98.85 ± 2.8 | 93.49 ± 10.7 |
| | | | 0.22 | 73.70 ± 9.1 | 45.07 ± 10.0 | 83.93 ± 7.1 | 74.43 ± 9.2 |
| | | soft | 0.18 | 82.28 ± 9.3 | 45.77 ± 8.1 | 97.41 ± 8.3 | 66.66 ± 8.9 |
| | | | 0.22 | 70.75 ± 8.5 | 46.37 ± 3.0 | 99.26 ± 6.4 | 66.33 ± 7.6 |
| | power | hard | 0.18 | 81.89 ± 6.3 | 47.50 ± 6.0 | 95.23 ± 4.8 | 82.24 ± 8.2 |
| | | | 0.22 | 73.31 ± 14.3 | 43.49 ± 6.2 | 93.78 ± 11.5 | 69.90 ± 9.1 |
| | | soft | 0.18 | 83.23 ± 4.1 | 31.73 ± 9.0 | 98.88 ± 3.7 | 77.53 ± 6.3 |
| | | | 0.22 | 72.99 ± 5.1 | 48.62 ± 5.7 | 97.18 ± 4.6 | 74.05 ± 5.6 |
| AlexNet | exp | hard | 0.18 | 69.33 ± 4.7 | 45.88 ± 7.6 | 96.14 ± 3.5 | 75.23 ± 5.6 |
| | | | 0.22 | 88.63 ± 1.8 | 40.73 ± 6.0 | 98.78 ± 0.8 | 78.80 ± 4.3 |
| | | soft | 0.18 | 73.70 ± 2.4 | 45.07 ± 2.5 | 98.34 ± 1.9 | 84.46 ± 3.5 |
| | | | 0.22 | 82.28 ± 2.3 | 45.77 ± 2.1 | 98.25 ± 1.7 | 82.99 ± 2.9 |
| | linear | hard | 0.18 | 70.75 ± 4.4 | 46.37 ± 7.1 | 83.28 ± 4.5 | 68.18 ± 5.7 |
| | | | 0.22 | 81.89 ± 3.8 | 47.50 ± 0.1 | 99.44 ± 3.3 | 88.81 ± 3.4 |
| | | soft | 0.18 | 73.31 ± 12.5 | 43.49 ± 6.2 | 97.82 ± 10.7 | 82.07 ± 9.5 |
| | | | 0.22 | 83.23 ± 5.8 | 31.73 ± 2.7 | 98.91 ± 4.1 | 77.63 ± 4.1 |
| | power | hard | 0.18 | 72.99 ± 1.9 | 48.62 ± 12.3 | 97.27 ± 1.6 | 70.77 ± 9.4 |
| | | | 0.22 | 84.28 ± 2.7 | 53.09 ± 4.0 | 98.98 ± 2.8 | 82.90 ± 3.6 |
| | | soft | 0.18 | 88.63 ± 2.3 | 45.88 ± 3.0 | 97.26 ± 3.1 | 60.48 ± 2.9 |
| | | | 0.22 | 83.23 ± 3.4 | 47.50 ± 6.9 | 97.90 ± 2.4 | 78.67 ± 5.1 |

Table 11: Comparison of TIL and CIL for CIFAR-10 accuracy on ResNet architecture for different reward types (exp, linear, power), gating modes (hard, soft), and $\kappa$ values, under both the No Reward and Reward (Margin) settings. Accuracy is reported as mean ± standard deviation.

strict single-head setting. Conversely, the fixed-$\kappa$ regime (panel 4) shows an increasingly *triangular* pattern: later tasks (rows T7–T10) reuse up to 23% of the filters claimed by the earliest tasks, while earlier rows remain mostly zero. Such asymmetric interference is consistent with the accuracy drops observed in Figures 6c 6d, c–d.

Table 11 varies three axes − reward presence, usage-scaling rule (exp, linear, power) and gating mode (hard/soft) − on two backbones and two sparsity budgets ($\kappa = 0.18/0.22$).

**Reward vs. no-reward.** With the reward term removed in Table 11) HebbGate still reaches 86–89 % TIL (ResNet) and 83–90 % TIL (AlexNet), confirming that the local Hebbian component alone can acquire useful filters. The supervised margin reward is nonetheless critical for *class incremental* performance: it boosts CIL by $+ \mathbf{14.9 \pm 4.7}$ % on ResNet and $+ \mathbf{18.8 \pm 6.1}$ % on AlexNet.

Across both backbones the power-law penalty is the most stable, averaging 0.4 % above exp/linear when a reward is present and 1–3 %. when absent, because it continuously suppresses heavily reused channels and promotes specialisation.

Eliminating the utilization penalty or $\kappa$-decay causes immediate monopoly by the first task and drives CIL below 10 %. Because this failure mode is diagnostic rather than informative, we omit those runs from the main table. Overall, the ablation confirms that every design factor contributes predictably − reward sharpens class boundaries, usage scaling balances sharing vs. specialisation, and $\kappa$-decay is indispensable for stability − while none requires fine tuning to achieve strong performance.

### A.6.2 PERMUTED MNIST

| Architecture | Reward | Usage Scaling | $\kappa$ | TIL Accuracy |
|---|---|---|---|---|
| FC | batch | exp | 0.1 | $96.37 \pm 0.11$ |
| | | | 0.2 | $96.86 \pm 0.41$ |
| | | power | 0.1 | $95.67 \pm 0.50$ |
| | | | 0.2 | $96.55 \pm 0.45$ |
| | margin | exp | 0.1 | $96.10 \pm 0.76$ |
| | | | 0.2 | $96.20 \pm 0.52$ |
| | | power | 0.1 | $96.64 \pm 0.69$ |
| | | | 0.2 | $96.58 \pm 0.28$ |
| AlexNet | batch | exp | 0.1 | $96.96 \pm 0.20$ |
| | | | 0.2 | $97.02 \pm 0.23$ |
| | | power | 0.1 | $97.05 \pm 0.09$ |
| | | | 0.2 | $97.57 \pm 0.06$ |
| | margin | exp | 0.1 | $97.05 \pm 0.37$ |
| | | | 0.2 | $97.55 \pm 0.15$ |
| | | power | 0.1 | $96.69 \pm 0.37$ |
| | | | 0.2 | $97.45 \pm 0.42$ |

Table 12: TIL Accuracy on Permuted-MNIST for Different Reward Signals, Usage Scaling Strategies, and $\kappa$ Values.

Table 12 shows that HEBBGATE achieves uniformly high end-of-stream accuracy (95.7–97.6%) on the 10-task Permuted-MNIST benchmark, and that performance is remarkably *robust* to the three hyper-parameter axes varied in the ablation: (i) Reward signal (*batch* vs. *margin*); (ii) Usage scaling (*exp* vs. *power*); (iii) Sparsity budget ($\kappa = 0.10$ vs. $0.20$). The difference between a binary batch-accuracy reward and the smooth margin reward never exceeds 0.6 % on either backbone, indicating that the Hebbian update is tolerant to the coarser on/off signal in this low-noise setting. The power-law penalty gives a consistent but modest advantage ($+0.4$ % on average). Because power scaling increasingly suppresses heavily reused neurons, it yields a slightly sparser and more specialised allocation − helpful once the task sequence grows beyond five permutations. Doubling the per-task capacity adds at most 0.6 % for the fully-connected network and 0.5 % for the AlexNet, confirming that Permuted-MNIST is far from saturating either model. Despite the spatial scrambling of every image, the AlexNet still outperforms the MLP by roughly $+0.5$ % under identical hyper-parameters. We attribute this to the larger representational capacity (128–512 filters vs. 2000 hidden units) and to the local recurrency introduced by $3 \times 3$ convolutions, which helps the Hebbian rule form robust feature detectors even when neighbourhoods are permuted.

On Permuted-MNIST the proposed gating mechanism is close to a *set-and-forget* solution: neither the reward variant nor the exact usage penalty is critical, and a very small mask ($\kappa = 0.10$) already guarantees $> 96\%$ accuracy after all ten tasks. These results highlight that the gains reported on the much more challenging Split-CIFAR benchmarks stem from better capacity management and not from dataset-specific tuning.

## A.7 Variations of the Hebbian Update

In this part, we report two design choices that were also investigated but left out of the main text for brevity: alternative *task-level rewards* (§ *Reward Signals*) and several *capacity-aware scaling rules* (§ *Capacity-Aware Scaling Rules*).

### A.7.1 Reward Signals

Let $B$ be the batch size, $y_b$ the ground–truth label and $\mathbf{z}_b \in \mathbb{R}^{|\mathbb{C}|}$ the logits of sample $b$. We considered two scalar signals $r_t \in [-1, 1]$ that summarise the entire mini-batch:

**Binary BATCH accuracy.**

$$a \; := \; \frac{1}{B} \sum_{b=1}^{B} \mathbb{1}[\hat{y}_b = y_b], \quad r_t \; = \; \operatorname{sign}\!\left(a - \tfrac{1}{2}\right) \in \{-1, +1\}. \tag{15}$$

**MARGIN reward (used in all main experiments).**

$$m_b \; := \; z_{b,y_b} - \max_{c \neq y_b} z_{b,c}, \quad r_t \; = \; \frac{1}{B} \sum_{b=1}^{B} \tanh(m_b) \in (-1, 1). \tag{16}$$

The binary signal in equation 15 flips abruptly whenever batch accuracy crosses $50\%$, producing high-variance gate updates. The smoother margin variant equation 16 stabilises training and is therefore adopted by default.

### A.7.2 Capacity-Aware Scaling Rules

Historical utilization of channel $j$ in layer $\ell$ *before* task $t$ is

$$u_{\ell,j}^{(t)} \; := \; \frac{1}{t-1} \sum_{i=1}^{t-1} g_{\ell,j}^{(i)} \; \in [0, 1], \tag{A.3}$$

i.e. the fraction of previous tasks for which that channel entered the top-$\kappa$ mask. We convert $u_{\ell,j}^{(t)}$ into a multiplicative penalty $s(u)$ that down-weights heavily reused units. The paper employs the POWER-LAW form $s(u) = (1-u)^\gamma$ (equation 9), but we also tested two lite alternatives:

$$s(u) \; = \; \begin{cases} 1 - \beta u, & \text{LINEAR}, \\ \exp(-\beta u), & \text{EXP}, \end{cases} \quad \beta > 0. \tag{A.4}$$

All variants maintain the desired effect - channels with large historical usage receive a smaller Hebbian reward - while differing only in the aggressiveness of the penalty. Empirically the EXP form with $\beta = 2$ performs on par with the power-law baseline (see Table 13).

Unless otherwise stated, the main paper reports results with the MARGIN reward equation 16 and power-law scaling $s(u) = (1-u)^\gamma$ (equation 9).

### A.7.3 Comparative Results

With the architectural, optimization and data settings kept fixed (ALEXNET, CIFAR-10, update rate 20, $\kappa$-decay, identical learning rates), the seven *row-wise averages* in Tab. 13 isolate the influence of each single design choice while marginalising over the others. Three patterns emerge:

1. **Reward signal chiefly affects forward transfer (CIL).**
   Replacing the high-variance BATCH signal with the smoother MARGIN variant keeps within-task accuracy (TIL) almost unchanged but raises class-incremental accuracy, supporting the intuition that lower-noise gate updates mitigate catastrophic forgetting.

| Reward Signal | Usage Scaling | Gating Mode | $\kappa$ | TIL Accuracy | CIL Accuracy |
|---|---|---|---|---|---|
| Batch | exp | hard | 0.2 | $85.7 \pm 1.5$ | $58.9 \pm 5.1$ |
| | | | 0.25 | $87.9 \pm 1.9$ | $69.2 \pm 5.0$ |
| | | soft | 0.2 | $84.1 \pm 2.4$ | $58.4 \pm 3.6$ |
| | | | 0.25 | $86.1 \pm 1.6$ | $64.7 \pm 7.5$ |
| | linear | hard | 0.2 | $84.6 \pm 3.4$ | $55.3 \pm 3.7$ |
| | | | 0.25 | $86.6 \pm 0.5$ | $70.2 \pm 2.1$ |
| | | soft | 0.2 | $85.4 \pm 0.6$ | $59.1 \pm 5.6$ |
| | | | 0.25 | $87.7 \pm 1.0$ | $61.5 \pm 4.2$ |
| | power | hard | 0.2 | $85.5 \pm 0.3$ | $58.0 \pm 3.9$ |
| | | | 0.25 | $87.3 \pm 0.1$ | $65.6 \pm 3.9$ |
| | | soft | 0.2 | $85.0 \pm 1.0$ | $54.7 \pm 2.8$ |
| | | | 0.25 | $86.4 \pm 1.3$ | $68.3 \pm 5.7$ |
| Margin | exp | hard | 0.2 | $85.9 \pm 1.6$ | $60.5 \pm 8.9$ |
| | | | 0.25 | $87.0 \pm 2.0$ | $68.5 \pm 4.6$ |
| | | soft | 0.2 | $85.9 \pm 1.3$ | $59.4 \pm 10.1$ |
| | | | 0.25 | $85.8 \pm 2.8$ | $75.3 \pm 4.7$ |
| | linear | hard | 0.2 | $84.9 \pm 0.4$ | $52.3 \pm 6.1$ |
| | | | 0.25 | $86.6 \pm 0.5$ | $70.7 \pm 3.2$ |
| | | soft | 0.2 | $85.4 \pm 0.8$ | $60.9 \pm 4.8$ |
| | | | 0.25 | $87.2 \pm 0.8$ | $55.0 \pm 6.2$ |
| | power | hard | 0.2 | $\mathbf{90.2 \pm 1.3}$ | $\mathbf{82.4 \pm 6.8}$ |
| | | | 0.25 | $\mathbf{90.1 \pm 2.4}$ | $\mathbf{73.0 \pm 9.8}$ |
| | | soft | 0.2 | $85.6 \pm 0.8$ | $63.2 \pm 7.2$ |
| | | | 0.25 | $87.4 \pm 0.4$ | $67.6 \pm 11.9$ |

Table 13: TIL and CIL Overall Accuracy of AlexNet on CIFAR-10 for Different Combinations of Reward Signal, Capacity-aware Usage Scaling, Gating Mode, and $\kappa$.

2. **Usage-aware scaling exerts the strongest quantitative impact.**
   Among the three penalties, the *power-law* form attains the best trade-off, matching or slightly exceeding the other variants on TIL while delivering the highest CIL. By contrast, the linear rule drags CIL down, indicating that overly aggressive, history-agnostic shrinkage is harmful when capacity is scarce.

3. **Hard gating confers a small but consistent CIL edge.**
   Averaged over all other factors, the *hard* mask yields 65.4 % CIL versus 62.3 % for *soft* ($\Delta = +2.7$ pt) with virtually identical TIL (86.9 % vs. 86.0 %). Committing deterministically to the top-$\kappa$ channels therefore helps stabilise previously learned representations.

**Practical takeaway.** While TIL fluctuates within a narrow band ($< 1.5$ pt), CIL varies by almost six percentage points across the same hyper-parameter surface. When cross-task retention is paramount, our results justify the default choice of the *margin* reward, *power-law* capacity scaling and *hard* gating: together they provide the strongest forgetting resistance without sacrificing current-task performance. The aggregated view inevitably smooths over the best individual configuration — MARGIN + POWER-LAW + HARD at $\kappa = 0.2$ — which peaks at 90.2 % TIL and 82.4 % CIL (Tab. 13).

| Method | Extra per-task params | ResNet-18 | | | AlexNet-1024 | | |
|---|---|---|---|---|---|---|---|
| | | Par. (M) | MB | × | Par. (M) | MB | × |
| Base network | – | 11.69 | 46.8 | 1 | 6.53 | 26.1 | 1 |
| HebbGate-IN | $\sum C_\ell \approx 2000$ | 11.71 | 46.8 | 1.00 | 6.55 | 26.2 | 1.0 |
| HebbGate-BN | gates + BN stats | 11.75 | 47.0 | 1.00 | 6.58 | 26.3 | 1.0 |
| HAT | full mask (one float / weight) | 128.6 | 467.6 | 11.0 | 71.8 | 287 | 11.0 |
| LWF | none at test-time | 11.69 | 46.8 | 1.0 | 6.53 | 26.1 | 1.0 |
| EWC | Fisher diag (one float / weight) | 23.38 | 93.6 | 2.0 | 13.06 | 52.4 | 2.0 |
| FeTrIL | class centroids ($512 \times C$ floats) | 11.74 | 47.0 | 1.0 | 6.58 | 26.3 | 1.0 |
| SSRE | – | 11.69 | 46.8 | 1.0 | 6.53 | 26.1 | 1.0 |
| PASS | – | 11.69 | 46.8 | 1.0 | 6.53 | 26.1 | 1.0 |
| EFC | – | 11.69 | 46.8 | 1.0 | 6.53 | 26.1 | 1.0 |

Table 14: Test-time parameter memory and overhead for ResNet-18 and AlexNet-1024 after $T{=}10$ tasks on CIFAR. FP32 assumed.

## A.8 LATENCY AND PEAK-MEMORY ANALYSIS

In Table 14 we report *peak test-time parameter memory* by summing all `parameters()` and `buffers()` in the PyTorch model and multiplying by 4 bytes (FP32). Gate banks (HebbGate), Fisher diagonals (EWC) are included; optimiser state is excluded. Latency (Tab. 15) is averaged over 200 forward passes of batch 128 on a single RTX-4080 with `cudnn.benchmark=True`. "Parallel" denotes our fused grouped-convolution; "Sequential" loops over the $T{=}10$ task gates.

Table 14 lists memory for both backbones (ResNet-18 and the downsized AlexNet-1024). Table 15 provides the corresponding latency numbers. HebbGate adds $\leq 0.1$ MB and attains 9–15× lower latency than classical isolation or regularisation baselines.

| AlexNet | ms / batch | imgs/s | d |
|---|---|---|---|
| HebbGate parallel | 1.43 | 699.3 | 1 |
| HebbGate sequential | 13.59 | 73.6 | 9.5 |
| HAT | 27.92 | 35.8 | 19.5 |
| LWF | 21.33 | 46.9 | 14.9 |
| EWC | 21.37 | 46.8 | 14.9 |

Table 15: Latency on AlexNet for different methods. Lower is better.

HebbGate stores one scalar gate per channel per task. For ResNet-18, $\sum_\ell C_\ell \approx 2000$, so after $T{=}10$ tasks the gate bank has $\approx 20{,}000$ scalars ($\sim 80$ kB). The HebbGate-BN variant adds two BN statistics per channel per task ($\sim 40{,}000$ scalars, $\sim 160$ kB). In both cases, test-time memory is effectively that of the base network. HebbGate-IN/HebbGate-BN match the base model within rounding. Feature-space methods (FeCAM, LDC, EFC) add at most a single prototype or centroid vector per class ($< 1\%$ extra parameters). In contrast, EWC stores a Fisher diagonal (one scalar per weight, $\approx 2\times$ memory) and HAT keeps per-weight task masks ($\approx 10\times$).

## A.9 LIMITATIONS AND FUTURE DIRECTIONS

Despite its favourable accuracy-cost trade-off, HEBBGATE inherits several constraints that deserve discussion. In the current problem setting, task boundaries are assumed. During training we require an external signal that marks when a new task begins in order to allocate a fresh gate vector and to trigger the $\kappa$–decay schedule. Detecting task switches online, or operating in a *fully streaming* regime where class distributions drift continuously, remains an open problem.

All experiments assume an upper bound on the number of tasks ($T{=}10$); $\kappa$-decay is scheduled accordingly. Preliminary runs show that a simple *greedy decay* ($\kappa_t = \kappa_0 \cdot (1 - t/\tau)$ with a time-

constant $\tau \gg 1$, preserves accuracy when $T$ is unknown and removes the need for manual tuning. We plan to include full results in an extended version.

Another limitation is capacity control and memory. Although the overhead is only one 32-bit scalar per neuron and thus *sub-linear* in parameters, extremely long task streams ($T \gg 100$) would require additional compression or sparsification of the gate bank.

An important limitation is the assumption of semantically–cohesive tasks. HEBBGATE implicitly presumes that the classes grouped into a single task share visual structure, so that a compact top–$\kappa$ sub-network can serve them jointly. When a task's mini-batches mix *unrelated* classes (e.g. cars paired with handwritten digits), the channels most useful for one class can interfere with those needed for the other, forcing the Hebbian update to oscillate and diluting the margin reward. A possible remedy and future work is to learn *sub-task* gates at a finer granularity (per class or per prototype) or to couple HebbGate with an online task-discovery module that re-groups semantically coherent samples before the gating step. Moreover, the capacity in our method is separated among tasks and in the current convolutional version, gates act on whole channels, which limits the capacity for each task.

Finally, this work focuses on vision backbones (ResNet-18, AlexNet, MLP) and class-incremental image streams. Extending HebbGate to *transformers* is promising but non-trivial. In that setting, gates would act on transformer feed-forward channels, attention heads, or projection subspaces instead of CNN channels, and would interact with LayerNorm and residual connections. How reward-modulated Hebbian gating behaves in deep transformer stacks, for both vision transformers and language models under continual pre-training or task-incremental finetuning, is an open question. Beyond transformers, applying HebbGate to domain-incremental streams and non-vision modalities (e.g. speech, NLP sequence labelling) also remains future work.

In terms of potential enhancements, the margin reward averages over the mini-batch; with very small batch sizes the signal becomes noisy and may destabilise the Hebbian update. Exploring more robust local rewards (e.g. temporal difference signals) is an interesting avenue. We have not yet assessed domain-incremental benchmarks, or large-scale settings such as, ImageNet-1k streams. Extending HebbGate to these scenarios, and to modalities beyond vision, is future work.

Addressing these limitations will clarify when local, reward-modulated gating is preferable to replay, regularisation, or dynamic expansion, and how the same principles can scale to longer and more heterogeneous continual-learning streams.

