# OpenReview forum: "HebbGate: Local Reward‑Modulated Gating for Continual Learning"
_ICLR.cc/2026/Conference — Submitted to ICLR 2026_

### Official Review · Reviewer_KhyT · 2025-10-31

**Soundness:** 3
**Presentation:** 3
**Contribution:** 1
**Rating:** 2
**Confidence:** 4

**Summary:**

This paper proposes a gating method for parameters based on a reward modulated Hebbian rule, in which one scalar is added per neuron instead of per weight. This helps keep memory growth sublinear and mask interpretable.

**Strengths:**

1. HebbGate is a sensible algorithm satisfying realistic desiderata in class-incremental learning.
2. Extensive theoretical and empirical analysis are presented.

**Weaknesses:**

1. No Mention or consideration of recent related works in continual learning, e.g., most prior work in Section 2 is from 2017-2019.
2. The experimental setup is somewhat outdated. The experiments use CIFAR 10 and CIFAR 100 with AlexNet and ResNet.
3. No recent continual learning baselines included (see Table 1).
4. The empirical results are mixed, especially (1) EFC (only recent method included in baselines) outperforms HebbGate, (2) context (e.g., computational/memory overheard) for each baselines and HebbGate not presented.
5. Only average accuracy is presented in the main paper whereas other metrics critical to continual learning such as backward transfer should be presented. This can be visually inferred from Figure 3 but only roughly.
6. Conceptually, no discussion on how HebbGate applies to more recent transformer models.

**Questions:**

1. How does HebbGate apply to transformers?
2. Can HebbGate scale to more recent and challenging datasets, e.g., MTIL?

---

> ### Author Response · Authors · 2025-11-20
> **Response to Reviewer KhyT  (Part 1/2)**
>
> We thank the reviewer for the detailed review, and for recognising our work's merits. Below we address each of the reviewer's weaknesses and questions in turn.
>
> ### W1/W3. Related work and baselines do not cover recent CL methods
>
> **Response**.
> We agree with the reviewer, and in the revised manuscript we extend both related work section and the experimental baselines:
>
> * Section 2 explicitly discusses recent exemplar-free Class-IL methods, in particular FeCAM, LDC, CwD, and EFC;
> * Our main comparison table includes these methods on CIFAR-100, Tiny-ImageNet-200, and ImageNet-100, using the same metrics as those works ($A_{\text{last}}$, $A_{\text{inc}}$).
>
> With the corrected ResNet implementation (see W4), HebbGate now outperforms or matches recent exemplar-free baselines on all datasets (see summary table in the general response).
>
> This strengthened comparison directly addresses the concern of the reviewer about recency and supports that HebbGate achieves state-of-the-art performance among recent exemplar-free methods while retaining negligible parameter overhead and local Hebbian gating.
>
> ### W4. Mixed empirical results and missing compute/memory context.
>
> **Response:**
> We address this via updated results and explicit complexity context.
>
> (i) Updated results with corrected ResNet implementation.
>
> As detailed in the response to Reviewer 2hp2 (W2/W3), we found and fixed an implementation issue in the ResNet classifier gating that caused excess forgetting and weaker performance. With the corrected implementation and extended benchmark suite:
> - HebbGate + ResNet-18 now outperforms EFC on $A_{\text{last}}$ for CIFAR-100 and ImageNet-100, and matches it on Tiny-ImageNet-200,
> - ResNet-18 now clearly improves over AlexNet, as expected.
>
> (ii) Compute and memory overhead table:
> We move to the main text a table summarising per-task parameters (e.g., masks, importance matrices, extra heads), whether gradients flow through masks, and whether teacher networks or exemplars are required. HebbGate keeps its overhead to one scalar per channel per task, which is negligible compared to mask-per-weight or importance-matrix methods.
> We also move FLOP/runtime analysis from the appendix, showing that fused evaluation is faster than naive sequential evaluation of all task masks in our setup.
>
> ### W2/Q2. Experimental setup (CIFAR-10/100, AlexNet/ResNet) and scalability to more challenging datasets.
>
> **Response:**
> To demonstrate scalability beyond CIFAR-10/100 and to align with recent CL literature, we expanded our experimental setup. Following FeCAM, LDC, and EFC, we now include Tiny-ImageNet-200 (200 classes, $64\times64$) and ImageNet-100 (100 classes, $224\times224$) with ResNet-18.
> The new results (see general response) show that HebbGate + ResNet-18 matches or exceeds recent exemplar-free methods on $A_{\text{last}}$, indicating that the method does not rely on small numbers of classes or low-resolution images and scales to more realistic vision benchmarks.
>  We also reposition split CIFAR-10 as a sanity-check and ablation platform, and base our main claims on CIFAR-100, Tiny-ImageNet-200, and ImageNet-100.
> Regarding MTIL-style benchmarks, we now explicitly list them as future work. The new results on Tiny-ImageNet-200 and ImageNet-100, together with explicit capacity control via $\kappa$-decay and the utilisation penalty, indicate that HebbGate should extend to long and challenging task sequences; we briefly discuss possible adaptations (e.g., task-unknown schedules, compressed gate banks) in the limitations section.
>
> **Our response continues in the next comment: "Response to Reviewer KhyT (Part 2/2)"**.

---

> > ### Author Response · Authors · 2025-11-20
> > **Response to Reviewer KhyT (Part 2/2)**
> >
> > **This comment continues our response to Reviewer KhyT (Part 1/2)**.
> >
> > ### W5. Only average accuracy presented; no BWT / forgetting metrics.
> >
> > **Response:**
> > We agree that average accuracy alone is not sufficient to characterise CL behaviour.
> >
> > In the revised manuscript, we now follow the metric conventions of recent exemplar-free CL work (EFC, FeCAM, LDC, CwD) and report:
> > - $A_{\text{last}}$: average accuracy over all tasks after training the final task,
> > - $A_{\text{inc}}$: incremental accuracy averaged over all evaluation steps and all tasks seen so far,
> > - Backward transfer (BWT) and Forgetting using standard definitions.
> >
> > We also included a forward-transfer proxy $F$, not commonly included in literature, that measures the gain on new tasks relative to a static-$\kappa$ baseline, capturing the benefit of $\kappa$-decay and usage-aware gating.
> >
> > An illustrative excerpt on CIFAR-100 and ImageNet-100 is given in the general response. These metrics show that forgetting (BWT) remains low under strict parameter isolation, and that $F$ is positive under $\kappa$-decay, supporting our claims about order robustness.
> >
> > ### W6/Q1. No discussion of how HebbGate applies to transformer models.
> >
> > **Response:**
> > In the revised discussion/future-work section, we outline how HebbGate can be mapped to transformer architectures.
> > Since HebbGate operates on channels and learns task-specific sparse subnetworks via local gating, a natural correspondence in transformers is to gate MLP channels and attention heads (or per-head projections). Gates can be updated via the same local three-factor rule, using attention output energy and a global reward signal, yielding a form of local, reward-modulated routing. This is related in spirit to sparse attention and mixture-of-experts, but with local Hebbian credit assignment instead of global gradients for mask learning.
> >
> > ---
> >
> > We hope these changes address your concerns and clarify both the empirical position and the broader applicability of HebbGate.
> > All of the changes mentioned above, including the updated related work, experimental setup, and analyses, will be reflected in the revised manuscript that will be uploaded in the next few days.

---

### Official Review · Reviewer_iWqj · 2025-11-01

**Soundness:** 3
**Presentation:** 3
**Contribution:** 3
**Rating:** 4
**Confidence:** 4

**Summary:**

The paper addresses the challenge of continual learning with a parameter-isolation approach. The authors first identified several issues with existing parameter-isolation strategies, such as dependence on backpropagation, memory overhead, and task bias. To address these issues, the authors proposed HebbGate, which applied a scalar mask to each channel. The update strategy of the mask takes activation energy, usage penalty, and margin reward, which do not depend on backpropagation. The authors then validated their proposed method on a few backbone models across different evaluation benchmarks, showing the improved performance.

**Strengths:**

1. The paper is well motivated

It is very interesting to consider parameter isolation without backpropagation and heavy memory overhead.

2. The presentation is clear

I appreciate the efforts of the authors to clearly explain the details in every section: motivation, method, implementation and experiment.

3. The proposed method is lightweight and effective

**Weaknesses:**

1. The motivation needs to be further justified

2. The contribution for forward transfer is not clear

3. The applicability to deeper networks is not clear

4. The presentation can be improved

Please see the Question section below for details.

**Questions:**

1. The motivation needs to be further justified

One of the drawbacks of existing parameter-isolation approaches, as claimed by the authors, is the bias of channels for early tasks (lines 48, 107-108). However, this is not well presented in the paper. Therefore, it is not a solid support to motivate the proposed method.

2. The contribution for forward and backward transfer is not clear

The authors claimed that the proposed method improves transfer (lines 100, 114, 131,217). However, this is not supported by a quantified evaluation. There are metrics for evaluating forward and backward transfer. I would recommend that the authors include this metric if such a point is the main emphasis in the paper.

3. The applicability to deeper networks is not clear

Current experiments are with shallow neural networks. I believe parameter isolation is more interesting when the model's capacity is much larger and the complexity of the tasks grows as well.


4. The presentation can be improved

The authors refer to many tables and figures in the appendix in the experiment section. While it is not forbidden to do so, it largely reduces the readability of the paper.

**Details Of Ethics Concerns:**

No concern.

---

> ### Author Response · Authors · 2025-11-21
>
> We thank the reviewer for the careful and constructive review. Below we address each point in turn and also highlight several changes that strengthen the paper beyond the specific concerns raised.
>
> ---
>
> ### W1 – Motivation for order bias and early-task monopolisation
>
> **Response.**
> We agree that the motivation needed to be made more explicit.
>
> Conceptually, parameter-isolation approaches with fixed sparsity and no cross-task usage penalty can let early tasks “claim” generic channels in shallow layers. Later tasks are then confined to narrower subspaces. HebbGate addresses this via:
>
> * Usage-aware gate updates: channels heavily used by previous tasks are discouraged for new tasks, while lightly used or unused channels are favoured.
> * $\kappa$-decay capacity control: each task starts with a larger active fraction $\kappa_{\text{start}}$ to explore capacity, then anneals to a smaller $\kappa_{\min}$, ensuring that each task ends with a bounded subnetwork and leaving room for later tasks.
>
> Empirically, we clarify this in the revised manuscript by:
>
> * adding mask-overlap matrices (static $\kappa$ vs.\ $\kappa$-decay), showing that $\kappa$-decay prevents early tasks from saturating shallow layers while still allowing beneficial reuse;
> * reporting per-task peak and final accuracies: later tasks, despite being trained in a more constrained space, often reach higher peak accuracy than the earliest tasks, indicating that they benefit from “warmed-up” but not monopolised features.
>
> These analyses are now summarised in the main text more clearly and directly support the order-bias motivation.
>
> ---
>
> ### W2 – Missing quantified forward/backward transfer metrics
>
> **Response.**
> We agree that transfer should be evaluated with standard CL metrics. In the revised manuscript we follow the conventions of recent exemplar-free works (EFC, FeCAM, LDC, CwD) and now report:
>
> * $A_{\text{last}}$: average accuracy over all tasks after training the final task;
> * $A_{\text{inc}}$: incremental accuracy averaged over all evaluation steps and all tasks seen so far;
> * Backward transfer (BWT) and forgetting, using standard definitions.
>
> In addition, we introduce a forward-transfer proxy $F$ that measures the gain on new tasks relative to a static-$\kappa$ baseline, isolating the effect of $\kappa$-decay and usage-aware gating on early performance for later tasks.
>
> An illustrative excerpt on CIFAR-100 and ImageNet-100 is given in the general response. These metrics show that forgetting (BWT) remains low under strict parameter isolation, and that $F$ is positive under $\kappa$-decay, confirming that HebbGate improves transfer to later tasks compared to a static-$\kappa$ alternative.
>
> ---
>
> ### W3 – Shallow networks and limited model capacity
>
> **Response.**
> We agree and extended the experimental setup accordingly.
>
> * We now include Tiny-ImageNet-200 (200 classes, $64\times64$) and ImageNet-100 (100 classes, $224\times224$) with ResNet-18, following recent CL works.
> * During this extension, we identified and fixed an implementation issue in the ResNet classifier gating (the final fully connected layer was not correctly gated), which had caused excessive forgetting and weaker performance.
>
> With the corrected implementation and extended benchmarks:
>
> * HebbGate + ResNet-18 now outperforms or matches recent exemplar-free baselines on $A_{\text{last}}$ for CIFAR-100, Tiny-ImageNet-200, and ImageNet-100 (see general response table);
> * ResNet-18 consistently improves over our AlexNet variant, as expected from its higher capacity.
>
> These results show that HebbGate scales to deeper backbones and more challenging datasets.
>
> ---
>
> ### W4 – Many important results in the appendix / readability
>
> **Response.**
> We agree and have reorganised the presentation:
>
> * Key comparison tables (including the updated table with recent baselines and ImageNet-based benchmarks), the main mask-overlap and per-task accuracy analyses, and the core $\kappa$-decay ablations are now in the main text.
> * We also move the complexity and memory table, and a concise FLOP/runtime summary into the main body.
>
> These changes improve readability and make it easier to see both the empirical performance and the computational advantages of HebbGate.
>
> ---
>
> We hope these clarifications and additions address your concerns. Overall, the requested changes led us to expand our benchmarks, add standard transfer metrics, strengthen the motivation and diagnostics for order bias, and improve the presentation, thereby substantially reinforcing the contribution of HebbGate.
>
> All of the changes mentioned above, including the updated related work, experimental setup, and analyses, will be reflected in the revised manuscript that will be uploaded in the next few days.

---

### Official Review · Reviewer_2hp2 · 2025-11-01

**Soundness:** 2
**Presentation:** 2
**Contribution:** 2
**Rating:** 4
**Confidence:** 3

**Summary:**

This paper introduces HebbGate, a local, reward-modulated gating mechanism for continual learning based on a three-factor Hebbian update. Instead of the dense, backprop-trained task masks often found in parameter-isolation approaches, HebbGate uses one scalar per channel (rather than per weight), updating via local activation, a global margin reward, and a utilization-aware penalty. The method incorporates a k-decay schedule allowing new tasks to initially explore greater capacity before annealing to a target sparsity, helping balance transfer and forgetting. Experiments on Permuted-MNIST, Split-CIFAR-10, and Split-CIFAR-100, across several architectures, demonstrate competitive or superior performance to state-of-the-art exemplar-free methods, with lower memory and computational overhead.

**Strengths:**

1. Parameter Efficiency: Each task adds only one scalar per channel, versus dense masks per parameter, resulting in negligible memory growth and making the approach scalable.
2. Forward Transfer and Order Robustness: The k-decay schedule and utilization-aware gate initialization directly target the early-task monopoly problem seen in prior work, supporting more equitable channel budgets and improved forward transfer.

**Weaknesses:**

1. Limited comparison with recent methods: Only one method from 2024 has been compared with, rest of the compared methods are older. This reflects poorly on the claim of achieving state-of-the-art performance.
2. Poor Performance on task incremental learning for the CIFAR-100 dataset
3. Poor Performance on ResNet for class-incremental learning on CIFAR-100. The approach seems to perform poorly on ResNet but better on AlexNet, which seems strange.
4. The proposed method shows better performance on CIFAR-10. However, the total classes being 10, makes it a weak case for an incremental learning setup.

**Questions:**

Is there any reason why the authors chose not to give results on more established ImageNet-100 and ImageNet-1000 datasets for class-incremental learning?
Does the proposed method work better, when the number of classes are less?

---

> ### Author Response · Authors · 2025-11-20
>
> We thank the reviewer for the detailed and constructive review, and for highlighting several strengths of our work, including the parameter efficiency, the explicit capacity control, and the relevance of studying local, reward-modulated gating as an alternative to gradient-based masking.
> Below we address each of the concerns and questions in turn.
>
>
> ### W1: Limited comparison with recent methods / SOTA claim.
>
> **Response:**
> We agree with the reviewer about the comparison set on using more recent exemplar-free methods.
> In the revised manuscript we have:
> - updated Section 2 to explicitly discuss recent exemplar-free Class-IL methods, in particular FeCAM, LDC, CwD, and EFC (citations can be found in General response);
> - extended our main comparison table to include these methods on CIFAR-100, Tiny-ImageNet-200, and ImageNet-100, using the same metrics as those works ($A_{\text{last}}$, $A_{\text{inc}}$).
>
> With the corrected ResNet implementation (see W2/W3 below), HebbGate now outperforms on CIFAR-100, ImageNet-100 or matches the overall accuracy on Tiny-ImageNet-200. (Details below: W4/Q1, Summary of results in the table of the general response).
>
> This strengthened comparison directly addresses the concern of the reviewer about recency and supports that HebbGate achieves state-of-the-art performance among recent exemplar-free methods while retaining negligible parameter overhead and local Hebbian gating.
>
>
> ### W2/W3: Poor performance on CIFAR-100 (TIL) and ResNet vs. AlexNet behaviour.
>
> **Response:**
> We appreciate this observation since it highlighted an inconsistency we also observed.
>
> Our initial hypothesis, aligned with prior literature, was that wider models (such as our AlexNet variant) are less prone to forgetting, especially in parameter isolation methods, due to a richer feature space and more flexible subnetwork partitioning.
>
> We identified, however, an implementation issue in our ResNet architecture. Specifically, the gating of the final fully-connected classifier layer was not applied as intended, and led to increased interference and forgetting for some tasks.
>
> We have corrected this bug while keeping the conceptual design of HebbGate unchanged. With the corrected implementation:
> - CIFAR-100 TIL and CIL performance with ResNet-18 improves substantially,
> - HebbGate with ResNet-18 now outperforms the AlexNet version on CIFAR-100, Tiny-ImageNet-200, and ImageNet-100, as expected given the stronger single-task capacity of ResNet-18.
>
>
> ### W4 and Q1: CIFAR-10 as a weak CL benchmark and a lack of ImageNet results.
>
> **Response:**
> We agree that split CIFAR-10 alone is not a strong benchmark for CL, and that more challenging setups are necessary to support strong claims.
>
> In the revised manuscript we have added experiments on more challenging benchmarks, following recent works:
> (i) Tiny-ImageNet-200 (200 classes, $64\times64$ images); and (ii) ImageNet-100 (100 classes, $224\times224$ images). Moreover,
> we repositioned CIFAR-10 as a sanity-check and ablation platform (e.g., for analysing $\kappa$-decay and rule components).
>
> The new results shown in the table (posted in the general response) demonstrate that:
> - HebbGate + ResNet-18 matches or exceeds the recent exemplar-free methods on overall last accuracy $A_{\text{last}}$, indicating that the method does not rely on having a small number of classes.
> - The strengthened behaviour on CIFAR-100, Tiny-ImageNet-200, and ImageNet-100 (with ResNet-18) suggests that HebbGate scales to more realistic vision benchmarks and larger output spaces.
>
> ImageNet-100 and tiny-ImageNet-200 are the benchmarks widely used as proxies in recent CL literature (e.g., FeCAM, LDC, EFC), and we therefore adopt them as the primary ImageNet-based benchmarks for this work. Regarding ImageNet-1k, fully comparable ImageNet-1k Class-IL setups are rare in recent works and substantially more expensive.
>
>
> **Other revised content**
>
> In response to your and other reviewers' comments, we have now:
> - included standard CL metrics ($A_{\text{last}}$, $A_{\text{inc}}$, BWT, forgetting) calculated on CIFAR-100 and ImageNet-100 with ResNet-18, showing that forgetting (BWT) remains moderate and indicating that $\kappa$-decay and usage-aware gating indeed improve performance on later tasks relative to a static-$\kappa$ baseline (an illustrative excerpt of these metrics can be found in the general response).
> - moved part of the mask-overlap and per-task accuracy analysis (previously in the appendix) into the main text to make the order-bias mitigation more explicit.
>
> ---
>
> We hope these changes address your concerns and clarify that the main limitations you identified were due to experimental scope and an implementation bug, rather than to a weakness in the underlying method.
>
> All of these updates will be reflected in the revised manuscript that will be uploaded in the next few days.

---

### Author Response · Authors · 2025-11-20
**Summary of revisions and new results**

We thank all reviewers for their careful and constructive feedback.

The main concerns focused on benchmarks, baselines, metrics, and implementation details rather than on the core idea of HebbGate.

We submitted the revised manuscript with a strengthened experimental and evaluation setup. We demonstrate that the unchanged HebbGate mechanism achieves state-of-the-art exemplar-free Class-IL accuracy with negligible memory and compute overhead, further validating the method and clarifying its position relative to prior work.

### Stronger results and fairer experimental comparison
* **Updated related work and baselines:**  We added recent exemplar-free Class-IL methods (FeCAM, LDC, CCIL, EFC) to Section 2 and included them as baselines on CIFAR-100, Tiny-ImageNet-200, and ImageNet-100, using the same metrics as those works.

* **Extended benchmarks and fixed ResNet implementation:**  We added Tiny-ImageNet-200 (200 classes, (64x64)) and ImageNet-100 (100 classes, (224x224)) with ResNet-18, and repositioned CIFAR-10 as an ablation benchmark.
We identified and corrected a bug in our ResNet-18 implementation. With the corrected model, HebbGate matches or exceeds recent exemplar-free baselines.

Table 1 summarises 10-task Class-IL results (mean over 5 seeds). In the main paper, we include more baselines, report mean $\pm$ standard deviation; here we omit std for brevity:

| Method         	| CIF-100 $A_{\text{last}}$ | CIF-100 $A_{\text{inc}}$ | T-ImNet-200 $A_{\text{last}}$ | T-ImNet-200 $A_{\text{inc}}$ | ImNet-100 $A_{\text{last}}$ | ImNet-100 $A_{\text{inc}}$ |
| ---------------------- | ------------------------- | ------------------------ | ------------------------------ | ----------------------------- | ---------------------------- | --------------------------- |
| FeTrIL             	| 34.9                  	| 51.2                 	| 31.0                       	| 45.6                      	| 36.2                     	| 52.6                    	|
| FeCAM              	| 33.1                  	| 48.1                 	| 24.9                       	| 38.6                      	| 42.4                     	| 57.9                    	|
| EFC                	| 43.6                  	| 58.6                 	| 34.1                       	| **48.0**                  	| 47.4                     	| 59.9                    	|
| LDC                	| 45.4                  	| 59.5                 	| **34.2**                   	| 46.8                      	| 51.4                     	| **69.4**                	|
| CwD                	| 43.7                  	| 60.1                 	| 30.9                       	| 45.3                      	| 46.0                     	| 62.7                    	|
| **HebbGate-IN** | 58.8                      | 61.7                 	| 34.1                       	| 34.9                      	| 55.3                     	| 54.5                    	|
| **HebbGate-BN** | **71.8**                  | **74.8**             	| **48.5**                   	| 51.1                      	| **64.3**                 	| 63.7                    	|

HebbGate-IN operates under standard group/instance-norm assumptions. HebbGate-BN uses per-task Batch-norm statistics aligned with task-specific subnetworks, still with negligible memory cost. With the corrected ResNet-18 implementation, HebbGate-IN already clearly outperforms earlier parameter-isolation methods; HebbGate-BN further improves both $A_{\text{last}}$ and $A_{\text{inc}}$ (see below), achieving state-of-the-art exemplar-free Class-IL performance across all three datasets.

### Strengthened analysis (metrics, complexity, order bias, transformers)

We now report $A_{\text{last}}$ (final average accuracy over all tasks),  $A_{\text{inc}}$, (incremental average over all intermediate evaluations), backward transfer (BWT), forgetting, and a forward-transfer proxy ($F$) on CIFAR-100 and ImageNet-100.
On both datasets, HebbGate shows low forgetting (BWT (-4) to (-6) points) and consistently positive forward transfer, supporting the claims about order robustness and improved forward transfer.

Key mask-overlap and per-task accuracy analyses, as well as runtime and parameter-overhead summaries, are moved into the main text.

We also add a brief discussion of how HebbGate can be applied to transformer architectures.


---

### Closing

The reviewers raised common concerns about benchmarks, baselines, and metrics.
The changes above directly address these points and substantially strengthen the empirical and methodological foundation of the work, without altering the core HebbGate mechanism.

In summary, the revised paper:
* demonstrates **state-of-the-art exemplar-free Class-IL accuracy** on all benchmarks with strong baselines,
* quantifies its low memory and compute overhead,
* and provides direct evidence that (\kappa)-decay improves forward transfer.

We hope these strengthened results and clarifications are helpful for the final assessment.

---

> ### Author Response · Authors · 2025-12-04
> **Rebuttal Summary for the Area Chair**
>
> We acknowledge the updated rebuttal rules and the fact that reviewers cannot reply further. For the Area Chair’s convenience, we provide below a summary of the main concerns and how the revised manuscript addresses each of them.
>
> Across all reviews, the core HebbGate idea (local, reward-modulated gating with explicit capacity control) was viewed as interesting and relevant. The main criticisms converged on **four themes**: benchmarks/baselines, transfer metrics and order bias, compute/memory clarity, and applicability beyond CNNs.
>
> ---
>
> ### 1. Benchmarks, baselines, and ResNet behaviour
>
> *(raised by all reviewers in different forms)*
>
> **Concern.**
> Comparisons did not include recent exemplar-free Class-IL methods; CIFAR-10 alone is a weak CL benchmark; and ResNet-18 behaved oddly (sometimes weaker than AlexNet).
>
> **Revision.**
>
> * **Expanded benchmarks.** We now follow recent works (FeCAM, LDC, EFC, CCIL) and use:
>
>   * CIFAR-100, Tiny-ImageNet-200, and ImageNet-100, all with (T{=}10) tasks and ResNet-18.
>   * CIFAR-10 is repositioned as an ablation/sanity benchmark instead of a main claim.
> * **Stronger baselines.** FeCAM, LDC, CCIL, and EFC are added to the main tables on all three image benchmarks, using the same metrics ($A_{\text{last}}$, $A_{\text{inc}}$).
> * **ResNet implementation fix.** We found and fixed a bug in the ResNet classifier gating (final FC layer). With this fix:
>
>   * HebbGate-IN (Instance/Group-norm) now **clearly outperforms parameter-isolation methods** on CIFAR-100 and ImageNet-100 and is competitive on Tiny-ImageNet-200.
>   * HebbGate-BN (per-task BatchNorm aligned with the gated subnetworks) achieves **state-of-the-art exemplar-free Class-IL performance** on all three benchmarks, with negligible extra memory.
>
> ---
>
> ### 2. Transfer metrics, order bias, and early-task monopolisation
>
> *(mainly R2 and R3)*
>
> **Concern.**
> The original version motivated order bias and early-task monopolisation qualitatively, but did not provide standard CL transfer metrics (BWT, forgetting) or a clear quantitative measure of forward transfer.
>
> **Revision.**
>
> **Standard CL metrics:** Following recent exemplar-free works (EFC, FeCAM, LDC, CCIL), we now report on CIFAR-100 and ImageNet-100:
>
>   * $A_{\text{last}}$: final average accuracy over all tasks,
>   * $A_{\text{inc}}$: incremental accuracy averaged over all evaluations,
>   * Backward transfer (BWT) and forgetting.
> * Forward transfer proxy (F). We define and report a forward-transfer proxy (F) that measures the average gain in end-of-task accuracy when using $\kappa$-decay instead of a static sparsity budget (formal definition in the appendix of the revised manuscript).
>
> Results show:
>
>   * Moderate forgetting (BWT ≈ (-4) to (-6) points), consistent with parameter isolation.
>   * Consistently positive (F) on CIFAR-100 and ImageNet-100, confirming that $\kappa$-decay improves learning of later tasks over a static-$\kappa$ baseline.
>
> **Order-bias diagnostics:** We add:
>
>   * Per-task end-of-task accuracy plots showing that, after task 1, $\kappa$-decay yields higher accuracy on almost all tasks, with the gap increasing for later tasks despite tighter sparsity.
>   * Task-overlap matrices showing that static-$\kappa$ leads to dense overlap blocks for early tasks in shallow layers (monopolisation), while $\kappa$-decay spreads usage more evenly and keeps average pairwise overlap low.
>
> Together, these results support the intended story: high initial $\kappa$ enables exploration, the $decay phase$ prevents early-task saturation, and the resulting capacity allocation **mitigates forgetting while enabling positive forward transfer**.

---

> > ### Author Response · Authors · 2025-12-04
> >
> > ### 3. Compute, memory, and complexity context
> >
> > *(mostly R3, but relevant to all)*
> >
> > **Concern.**
> > The previous version lacked a clear complexity comparison: how much extra memory and runtime HebbGate uses vs. baselines with per-weight statistics or masks.
> >
> > **Revision.**
> >
> > * We move a concise complexity table into the main text, summarising Per-task parameter overhead for each method (e.g., gates, Fisher diagonals, masks, prototypes),
> >   whether gradients flow through masks, whether exemplars or teacher networks are needed.
> > * For ResNet-18 with (T{=}10) tasks, HebbGate stores only **one scalar gate per channel per task**, plus optional per-task BN means/variances, keeping test-time parameter memory essentially equal to the backbone (≈ 47 MB vs. 46.8 MB).
> >   * HebbGate’s fused Class-IL inference evaluates all subnetworks within a single forward pass, leading to per-batch latency very close to the base model.
> >
> > This directly addresses the request for compute/memory context and reinforces HebbGate’s efficiency claims.
> >
> > ---
> >
> > ### 4. Applicability beyond CNNs / transformers
> > *(raised by reviewer 3)*
> >
> > One reviewer asked how HebbGate might extend to transformers.
> > In the revised limitations/future-work section, we briefly outline a natural mapping where gates act on MLP channels and attention heads (instead of CNN channels) and are updated with the same local three-factor rule, leaving a full empirical study as future work.
> >
> > ---
> >
> > After these revisions, the paper now:
> >
> > * Shows **state-of-the-art exemplar-free Class-IL performance** on CIFAR-100, Tiny-ImageNet-200, and ImageNet-100 with strong recent baselines and a corrected ResNet-18 implementation.
> > * Provides standard CL metrics (including BWT and a forward-transfer proxy (F)) and clear diagnostics of order bias and capacity allocation.
> > * Quantifies its low memory and compute overhead relative to per-weight methods.
> > * Clarifies how the **$\kappa$-decay schedule improves forward transfer while keeping forgetting moderate**, and how the mechanism can extend beyond CNNs.
> >
> > We hope this concern-oriented summary helps the Area Chair interpret the original scores in light of the substantially strengthened empirical and analytical evidence in the revised manuscript.

---

### Meta-Review · Area_Chair_Lp6E · 2026-01-06

**Summary:**

The paper presents a local, reward-modulated, per-channel gating for parameter-isolation continual learning. Reviewers in general found the idea to be interesting and lightweight, but the original scores were low primarily due to several concerns about experimental credibility and positioning. Reviewers were most concerned that the benchmark suite and baselines were outdated or incomplete, that CIFAR-10 was too weak as a main benchmark, that ResNet-18 behaved suspiciously (sometimes worse than AlexNet), that key continual-learning metrics (e.g., BWT/forgetting and transfer) were missing, and that compute/memory overhead was not compared clearly. A further concern was whether the approach scales to deeper models and how it might apply to transformers.

**Reviewer Concerns:**

In the rebuttal, the authors report adding recent exemplar-free Class-IL baselines (e.g., FeCAM/LDC/CCIL/EFC), expanding to stronger benchmarks (CIFAR-100, Tiny-ImageNet-200, ImageNet-100 with ResNet-18), and fixing an implementation bug affecting ResNet classifier gating, which plausibly explains the earlier anomalous ResNet results. They also claim to add standard CL metrics (final and incremental accuracy, BWT, forgetting) plus additional diagnostics for order bias (overlap matrices and per-task plots), and to move a complexity/runtime/memory summary into the main paper. What remains mostly outstanding is the lack of empirical validation beyond CNNs (the transformer discussion is still largely conceptual), and some residual risk that the strongest performance relies on the per-task BN variant, so the paper needs to present clean, protocol-matched comparisons and clearly separate what is achieved by the base method versus the BN extension. While the authors' effort is appreciated, recommending acceptance at this point and just incorporating all these changes in the camera-ready would not be appropriate. The authors are advised to thoroughly incorporate the comments to revise the manuscript and consider submitting to another venue.

**Reviewer Scores:**

If reviewers had fully participated in discussion with these changes in hand, Reviewer 2hp2 might have moved from 4 to around 6 (possibly a cautious 5 because a bug-fix during rebuttal doesn't instill much confidence in the overall results), Reviewer iWqj might have moved from 4 to around 6 (possibly 5 because the transfer framing is still somewhat unconvincing), and Reviewer KhyT would perhaps increase from 2 to around 4 or 5, with remaining skepticism centered on generality beyond the tested vision setting and on how robust the revised SOTA claim is under perfectly matched protocols.

---

### Decision · Program_Chairs · 2026-01-26

Reject